# TARTARUS: A Benchmarking Platform for Realistic And Practical Inverse Molecular Design

**AkshatKumar Nigam**[1], **Robert Pollice**[2,3,*], **Gary Tom**[3,4], **Kjell Jorner**[3],
**John Willes**[4], **Luca Thiede**[3], **Anshul Kundaje**[1], **and Alán Aspuru-Guzik**[3,4,5,6]

[1]Stanford University. [2]University of Groningen. [3]University of Toronto.
[4]Vector Institute. [5] Lebovic Fellow. [6] Canadian Institue for Advanced Research.
*r.pollice@rug.nl

## Abstract

The efficient exploration of chemical space to design molecules with intended properties enables the accelerated discovery of drugs, materials, and catalysts, and is one of the most important outstanding challenges in chemistry. Encouraged by the recent surge in computer power and artificial intelligence development, many algorithms have been developed to tackle this problem. However, despite the emergence of many new approaches in recent years, comparatively little progress has been made in developing realistic benchmarks that reflect the complexity of molecular design for real-world applications. In this work, we develop a set of practical benchmark tasks relying on physical simulation of molecular systems mimicking real-life molecular design problems for materials, drugs, and chemical reactions. Additionally, we demonstrate the utility and ease of use of our new benchmark set by demonstrating how to compare the performance of several well-established families of algorithms. Surprisingly, we find that model performance can strongly depend on the benchmark domain. We believe that our benchmark suite will help move the field towards more realistic molecular design benchmarks, and move the development of inverse molecular design algorithms closer to designing molecules that solve existing problems in both academia and industry alike.

## 1 Introduction

Inverse molecular design, which involves crafting molecules with specific optimal properties [1], poses a critical optimization challenge prevalent across various scientific fields. It holds particular significance in chemistry for designing drugs, catalysts, and materials. Each of these scenarios requires a complex balance between multiple desired properties for a realistic inverse design. Yet, the majority of comparisons between molecular design algorithms are primarily carried out on oversimplified tasks, thereby providing limited insights into their potential performance when confronting intricate design challenges commonplace in chemistry [2–4]. While simplistic and cost-effective benchmarks serve as vital tools during the initial stages of algorithm development, facilitating rapid testing and informed design choices, the lack of more realistic benchmarks obstructs their adoption by domain experts in chemistry or materials science. Our work aims to bridge this gap.

In this study, we propose a modular suite of design objectives directly inspired by problems regarding new materials, drugs, and chemical reactions. These problems rely on well-established simulation methods in computational chemistry, such as force fields, semi-empirical quantum chemistry, and density functional approximations. We test multiple popular molecular design algorithms on these objectives, offering an exemplary performance comparison. The design approaches selected represent some of the most important algorithm classes in the field. We also examine the impact of representations on the performance of some algorithms. Finally, we provide detailed instructions for

installation and setup for both the benchmarks and the molecular design algorithms, enabling the cheminformatics and machine learning communities to integrate them into their work and inspire the development of future methods. The core idea is to provide a unified benchmarking framework with a diverse selection of problem domains to assess the generalizability of inverse design algorithms. Notably, the scope of included benchmark domains is to be expanded upon in future work. Altogether, we believe this is a stepping stone towards the widespread adoption of molecular generative models for challenging design problems that permeate the chemical sciences.

## 2 Background

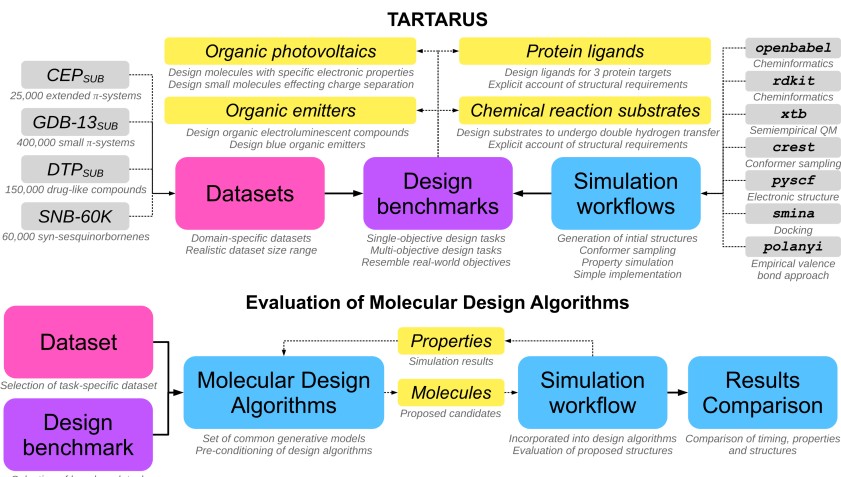

Figure 1: TARTARUS framework, highlighting its two core elements. *Top:* Real-world design tasks are defined and paired with simulation workflows and datasets. *Bottom:* The Evaluation Pipeline: Generative models are conditioned using reference datasets, sampled for structures, and evaluated based on their properties through custom simulation workflows. Model performance is assessed based on the alignment of the acquired properties with predefined objectives.

Generative models play a pivotal role in inverse molecular design, yet the progress of benchmark tasks for their performance assessment has been relatively slow [5, 6]. The penalized log P metric, frequently employed in current benchmarks, is characterized by its dependence on molecule size and chain composition, thus limiting its effectiveness [2, 7–9]. Numerous models have reached a saturation point on the task of generating molecules with maximum QED values [10–14]. The GuacaMol benchmark suite, with its array of goal-oriented objectives [3], often similarly results in near-perfect scores, thus obscuring useful comparisons [15–17]. This concern was traced back to unlimited property evaluations within the suite, with imposed limits revealing significant performance disparities and underscoring the importance of evaluation procedures explicitly accounting for evaluations [18]. The MOSES benchmark suite evaluates the ability of models to propose diverse compounds that mimic the distribution of training datasets [4]. However, the emergence of SELFIES and rudimentary algorithms have rendered these tasks trivial [19–21].

The recent trend of using molecular docking as a benchmark for generative models includes tasks like designing molecules complying with Lipinski's Rule of Five [22]. However, this often favors highly reactive and unstable molecules [7]. Additionally, molecular docking scores have recently become a popular metric [23–27, 22]. However, they were only recently incorporated into systematic benchmark platforms. In particular, the Therapeutics Data Commons (TDC) does provide a comprehensive benchmark platform, but crafting sensible, synthesizable, stable structures with low binding affinity estimates remains a formidable challenge [28]. The DOCKSTRING package supplies extensive datasets and benchmarks for a variety of machine learning strategies in drug design [29]. It constitutes an advance over previous methodologies by offering computational workflows and realistic molecular design benchmarks. However, the breadth of molecular design task extends beyond drug discovery. Accordingly, TARTARUS aspires to cover a more comprehensive set of molecular design problems, to include important structural domains previously neglected.

Molecular design algorithms have the potential to accelerate the discovery of materials for the benefit of humankind. However, they can just as well be used to design hazardous materials intentionally. While this pertains primarily to the molecular design algorithms themselves, benchmarking can also provide a contribution in that regard by identifying robust and generally applicable tools. However, first, we would like to emphasize that none of the molecular design benchmarks developed have a direct link to the development of potentially hazardous materials. Additionally, the molecular design algorithms employed in this work propose merely structures but do not provide any instructions for their synthesis making their malicious use unlikely.

This study introduces four molecular design benchmarks, each with a curated dataset of reference molecules with predicted properties, as depicted in Figure 1. The potential availability of reference data from laboratory experiments for at least subsets of our curated datasets was not considered. The goal is to present authentic benchmarks that aid in discovering structures optimized for specific target properties across diverse applications. To evaluate models with TARTARUS, we recommend training the generative model on the provided dataset, using 80% for training and 20% for hyperparameter optimization. The model should then propose structures for evaluation by the corresponding benchmark task. Structure optimization is initiated using the best reference molecule from the respective dataset. A constrained approach is followed, limiting the population size, iterations, and total proposed compounds to 5,000, with a capped runtime of 24 hours. Each optimization run is repeated five times for increased robustness and reproducibility. This resource-constrained comparison approach, we argue, is crucial for impartial method comparison and should be a community standard. The parameters and settings used for each model are detailed in the Supporting Information.

## 3 Results

In this section, we describe the molecular design benchmarks we developed, along with their reference datasets. We also present the results for all the generative models considered, namely: REINVENT [30], SMILES-VAE [31], SELFIES-VAE, MoFlow [32], SMILES-LSTM-HC [33, 3], SELFIES-LSTM-HC, GB-GA [34], and JANUS [7]. We would like to emphasize that this is far from a comprehensive list of all molecular design algorithms, but rather a small selection of various algorithmic approaches spanning the field. Moreover, the results provided heavily depend on both the model hyperparameters and the benchmark settings, such as available computing resources and number of property evaluations. For a comprehensive outline and discussion of the property simulation workflows developed in this work, we refer the reader to the Additional Results of the Supporting Information.

### 3.1 Design of Organic Photovoltaics

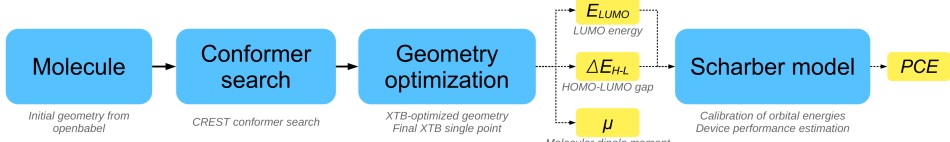

Figure 2: Overview of the property simulation workflow for the design of organic photovoltaics benchmarks. The code accepts a SMILES string, generates initial Cartesian coordinates with `Open Babel`, and performs conformer search and geometry optimization with `crest` and `xtb`. Finally, a single point calculation at the GFN2-xTB level of theory provides the HOMO and LUMO energies, the HOMO-LUMO gap and the molecular dipole moment. The power conversion efficiency (PCE) is computed from the simulated properties based on the Scharber model.

Our first benchmark objective set comprises two individual tasks inspired by organic photovoltaic (OPV) design [35]. The development of organic solar cells (OSCs) garners broad interest due to their potential to replace and expand upon the applications of predominant inorganic devices [36–39]. Key properties of photoactive materials for OSCs are their HOMO and LUMO energies (i.e., $E_{HOMO}$ and $E_{LUMO}$, respectively). Importantly, these properties of interest can be directly estimated using a standard ground state electronic structure simulation. Thus, we defined two benchmark objectives that are directly inspired by previous high-throughput virtual screening efforts in the literature aimed

at finding new photoactive materials for OPVs. The first task is based on the Harvard Clean Energy Project (CEP) [37, 40] and corresponds to the design of a small organic donor molecule to be used with [6,6]-phenyl-C61-butyric acid methyl ester (PCBM) as acceptor in a bulk heterojunction device [41, 42, 37, 40]. The second task is based on follow-up work of the CEP and corresponds to the design of a small organic acceptor molecule to be used in bulk heterojunction devices with poly[N-90-heptadecanyl-2,7-carbazole-alt-5,5-(40,70-di-2-thienyl-20,10,30-benzothiadiazole)] (PCDTBT) as a donor [43]. In both cases, the objective function is a combination of (1) the estimated power conversion efficiency (PCE) of the corresponding device, which is based on the Scharber model for single junction OSCs [41, 42] and derived from the results of an electronic structure calculation of the proposed molecule, and (2) the synthethic accessibility of the corresponding structure as quantified by the SAscore [44].

We also provide a reference dataset for training generative models. It is a subset of the Harvard Clean Energy Project Database (CEPDB) [40], originally encompassing 2.3 million organic compounds. Our subset, $CEP_{SUB}$, consists of approximately 25,000 molecules. We implemented a robust property simulation workflow for these benchmark tasks. This workflow accepts the SMILES string of proposed molecules as input [45, 46], generates an initial guess of the corresponding Cartesian coordinates, performs conformer sampling using the iMTD-GC workflow [47, 48] as implemented in `crest` [49] at the GFN-FF level of theory [50–52], and performs geometry optimization of the lowest energy conformer obtained at the GFN2-xTB level of theory [53, 54]. The predicted properties are taken from a final single point energy cal-

Table 1: Results for the organic photovoltaics benchmarks. Models are trained on a subset of the Harvard Clean Energy Project Database. Results are provided as mean and standard deviation of the best target objective values that are obtained in five independent runs in the form mean $\pm$ standard deviation. The top-performing molecule in the training set is also given ("Dataset"). Metrics: $PCE_{PCBM}$: PCBM power conversion efficiency; $PCE_{PCDTBT}$: PCDTBT power conversion efficiency; SAscore: synthetic accessibility score.

|  | $PCE_{PCBM} - SAscore$ | $PCE_{PCDTBT} - SAscore$ |
| --- | --- | --- |
| Dataset | 7.57 | 31.71 |
| SMILES-VAE | 7.44 $\pm$ 0.28 | 10.23 $\pm$ 11.14 |
| SELFIES-VAE | 7.05 $\pm$ 0.66 | 29.24 $\pm$ 0.65 |
| MoFlow | 7.08 $\pm$ 0.31 | 29.81 $\pm$ 0.37 |
| SMILES-LSTM-HC | 6.69 $\pm$ 0.40 | **31.79 $\pm$ 0.15** |
| SELFIES-LSTM-HC | 7.40 $\pm$ 0.41 | 30.71 $\pm$ 1.20 |
| REINVENT | 7.48 $\pm$ 0.11 | 30.47 $\pm$ 0.44 |
| GB-GA | **7.78 $\pm$ 0.02** | 30.24 $\pm$ 0.80 |
| JANUS | 7.59 $\pm$ 0.14 | 31.34 $\pm$ 0.74 |

culation of the converged geometry. An overview of the simulation workflow and the corresponding benchmark objectives is provided in Figure 2.

The results of the two benchmarks are summarized in Table 1. We observe that GB-GA shows best performance on the first benchmark task (7.78), while SMILES-LSTM-HC shows best performance on the second (31.79). While most models are capable to propose molecules with marginally improved PCEs, they are not able to both improve PCEs and diminish SAscores. When looking at the best molecules that the molecular design algorithms generated (cf. Supporting Information), it is interesting to see that the various models seem to produce structures with essentially equivalent core motifs. Additionally, it is apparent that, due to the inclusion of the SAscore in the design objectives, the proposed structures are more likely to be stable and synthesizable.

## 3.2 Design of Organic Emitters

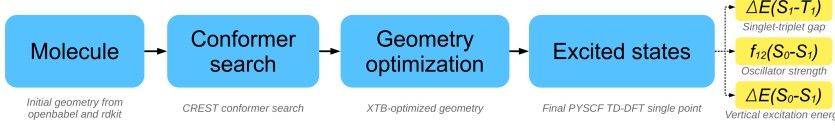

Figure 3: Overview of organic emitter design workflow. Starting with a SMILES string, the code executes conformer search and geometry optimization via `xtb`. TD-DFT single point calculation with `pyscf` extracts the singlet-triplet gap, oscillator strength, and vertical excitation energy.

The next set of benchmarks is inspired by the design of emissive materials for organic light-emitting diodes (OLEDs), which received significant attention in recent years [55–59] after the discovery of thermally activated delayed fluorescence (TADF) [60]. Their main applications are digital screens

and lighting devices [55]. Improving the efficiency of organic emissive materials relying on TADF can be achieved by minimization of their singlet-triplet gaps [61]. Additionally, fluorescence rates need to be increased which corresponds to maximizing the oscillator strength between the first excited singlet and the ground state [61]. Furthermore, blue OLEDs pose a challenge as they typically suffer from reduced internal quantum efficiencies and device lifetimes [61, 55]. Accordingly, we propose benchmarks targeting the design of emissive materials for OLEDs. The first task is to minimize their singlet-triplet gaps. The second task is to maximize their oscillator strengths. In a third objective, the goal is to combine the first two tasks and, additionally, keep the vertical excitation energies between the ground state and the first excited singlet state in the energy range of blue light. In these tasks, the SAscore of proposed molecules needs to be smaller than or equal to 4.5 for a high fitness.

For training the generative models, we selected a subset of GDB-13 [62], which originally consists of more than 970 million organic molecules with up to 13 non-hydrogen atoms, and extracted approximately 380,000 molecules, that consist to a high degree of organic molecules with extended planar conjugated systems of double bonds and lone pairs, i.e., $\pi$-systems, via a comprehensive set of filters (cf. Supporting Information). The property simulation workflow consists of conformer sampling using the iMTD-GC workflow [47, 48], as implemented in `crest` [49], at the GFN-FF level of theory [50–52], followed by geometry optimization of the lowest energy conformer obtained at the GFN0-xTB level of theory [53, 63, 64], and an excited state single point calculation via TD-DFT at the B3LYP/6-31G* level of theory [65–70] (Details in the Supporting Information). The entire simulation workflow is illustrated in Figure 3.

The performance of all the generative models employed on the set of organic emitter design benchmarks is summarized in Table 2 and compared directly to the best baseline fitness values extracted from the dataset. We observe that only JANUS, GB-GA, and the SELFIES-VAE successfully generate compounds that are comparable to or improve upon the best molecules in the training dataset and the lowest singlet-triplet gap with a value of 0.008 eV was found by JANUS. However, this value is only an insignificant improvement over the best reference structure (0.020 eV). These three models also achieved larger oscillator strengths than observed in the training dataset, with GB-GA achieving the highest average oscillator strength with a property value of 2.16. The SELFIES-VAE also achieved the highest fitness in the multi-objective benchmark task, with an average value of 0.17. All the remaining models were unable to outperform the best molecules in the dataset (-0.04). Finally, the best structures proposed by the generative models in these benchmark tasks (cf. Supporting Information) illustrate that some of them possess reactive structural moieties which is most likely due to the absence of an explicit inclusion of stability or synthesizability in the objective functions.

Table 2: Results for the organic emitters design benchmark objectives. Models are trained on a subset of the GDB-13 dataset. Results are provided as mean and standard deviation of the best objective values from five independent runs in the form mean $\pm$ standard deviation. The top-performing molecule in the training dataset is also included ("Dataset"). Metrics: $\Delta E(S_1 - T_1)$: singlet-triplet gap; $f_{12}$: $S_1$ and $S_0$ transition oscillator strength; $\Delta E(S_0\text{-}S_1)$: $S_0$ and $S_1$ vertical excitation energy.

| | $\Delta E(S_1 - T_1)$ | $f_{12}$ | $+f_{12} - \Delta E(S_1\text{-}T_1) - |\Delta E(S_0\text{-}S_1) - 3.2; eV|$ |
|---|---|---|---|
| Dataset | 0.020 | 2.97 | -0.04 |
| SMILES-VAE | $0.071 \pm 0.003$ | $0.50 \pm 0.27$ | $-0.57 \pm 0.33$ |
| SELFIES-VAE | $0.016 \pm 0.001$ | $0.36 \pm 0.31$ | $\mathbf{0.17 \pm 0.10}$ |
| MoFlow | $0.013 \pm 0.001$ | $0.81 \pm 0.11$ | $-0.04 \pm 0.06$ |
| SMILES-LSTM-HC | $0.015 \pm 0.002$ | $1.00 \pm 0.01$ | $-0.24 \pm 0.01$ |
| SELFIES-LSTM-HC | $0.013 \pm 0.003$ | $1.00 \pm 0.01$ | $-0.24 \pm 0.01$ |
| REINVENT | $0.014 \pm 0.003$ | $1.16 \pm 0.18$ | $-0.15 \pm 0.05$ |
| GB-GA | $\underline{0.012 \pm 0.002}$ | $\mathbf{2.14 \pm 0.45}$ | $\underline{0.07 \pm 0.03}$ |
| JANUS | $\mathbf{0.008 \pm 0.001}$ | $\underline{2.07 \pm 0.16}$ | $0.02 \pm 0.05$ |

## 3.3   Design of Protein Ligands

We also wanted to include molecular design objectives relevant for medicinal chemistry. In recent years, deep generative models have experienced a strong increase in popularity and adoption for drug design as they promise to accelerate discovery campaigns leading to significant cost reductions, and success stories were reported on in the literature [71–74]. Therefore, we decided to develop objectives that directly targets the design of ligands for specific proteins based on molecular docking simulations. We set up three benchmark tasks, each aimed at a different protein. As targets, we selected 1SYH,

Table 3: Performance metrics for protein-ligand design benchmarks, based on models trained on a subset of the DTP Open Compound Collection. Metrics show mean and standard deviation of optimal target objective values over five independent runs (mean ± standard deviation). "Dataset" and "Native Docking" values represent the top-performing molecule in the training dataset and the original ligands in their crystal structures, respectively. $\Delta E_X$ denotes docking score for protein target $X$, and SR reflects the success rate for molecules passing structural filters.

| | 1SYH | | | 6Y2F | | | 4LDE | | |
| | $\Delta E_{1SYH}$ | | SR | $\Delta E_{6Y2F}$ | | SR | $\Delta E_{4LDE}$ | | SR |
| | QuickVina2 | Smina | | QuickVina2 | Smina | | QuickVina2 | Smina | |
|---|---|---|---|---|---|---|---|---|---|
| Native Docking | -10.2 | -10.5 | 100.0% | -4.9 | -5.3 | 0.0% | -11.6 | -12.1 | 100.0% |
| Dataset | -9.9 | -10.2 | 100.0% | -8.2 | -8.2 | 100.0% | -12.2 | -13.1 | 100.0% |
| SMILES-VAE | -10.0 ± 0.7 | -10.4 ± 0.6 | 12.3% | -8.3 ± 0.6 | -8.9 ± 0.8 | 13.1% | -10.7 ± 0.2 | -11.1 ± 0.4 | 12.6% |
| SELFIES-VAE | -10.4 ± 0.3 | -10.9 ± 0.3 | 34.8% | -9.6 ± 0.5 | -10.1 ± 0.4 | 38.9% | -11.1 ± 0.4 | -11.9 ± 0.2 | 38.9% |
| MoFlow | -10.6 ± 0.4 | -11.0 ± 0.3 | 35.5% | -9.8 ± 0.4 | -10.6 ± 0.3 | 35.9% | -12.1 ± 0.4 | -13.0 ± 0.3 | 36.2% |
| SMILES-LSTM-HC | -10.6 ± 0.6 | -11.1 ± 0.4 | 71.7% | -10.0 ± 0.6 | -10.4 ± 0.7 | 70.1% | -12.4 ± 0.3 | -13.3 ± 0.4 | 73.9% |
| SELFIES-LSTM-HC | -10.8 ± 0.5 | -11.2 ± 0.2 | 73.2% | -10.4 ± 0.4 | -10.6 ± 0.6 | 71.5% | -12.7 ± 0.3 | -13.6 ± 0.5 | 75.6% |
| REINVENT | **-11.8 ± 0.4** | **-12.1 ± 0.2** | **77.8%** | -11.1 ± 0.3 | -11.4 ± 0.3 | 76.8% | -12.8 ± 0.2 | -13.7 ± 0.5 | **76.8%** |
| GB-GA | -11.6 ± 0.5 | -12.0 ± 0.2 | 72.6% | -10.9 ± 0.2 | -11.0 ± 0.2 | 73.9% | **-12.9 ± 0.1** | **-13.8 ± 0.4** | 71.4% |
| JANUS | -11.7 ± 0.4 | -11.9 ± 0.2 | 68.4% | **-11.3 ± 0.3** | **-11.9 ± 0.4** | 70.4% | -12.8 ± 0.2 | -13.6 ± 0.5 | 65.3% |

an ionotropic glutamate receptor associated with neurological and psychiatric diseases such as Alzheimer's, Parkinson's and epilepsy [75], 6Y2F, the main protease of SARS-CoV-2 responsible for translation of the viral RNA of the SARS-CoV-2 virus [76], and 4LDE, the $\beta$2-adrenoceptor GPCR receptor spanning the cell membrane and binding adrenaline, a hormone which mediates muscle relaxation and bronchodilation [77]. In all cases, crystal structures of the proteins bound to a ligand are available in the protein data bank (PDB) [78] and the objective is to minimize the docking score of a proposed molecule to one of these protein targets.

For training, we started from the Developmental Therapeutics Program (DTP) Open Compound Collection [79, 80], a set of about 250,000 molecules tested for treatment against cancer and the acquired immunodeficiency syndrome (AIDS) [81], and selected all structures passing the structure constraints leading to around 150,000 structures referred to as `DATASET`. Molecules included in this collection can be ordered for free, except for shipping costs, from the DTP [82]. The simulation workflow for these benchmark tasks is initiated with the SMILES string of the proposed molecule and creates an initial guess of the corresponding Cartesian coordinates with `openbabel`. Subsequently, we use `QuickVina2` [83] to introduce the newly proposed compound in the protein binding pocket, and perform a docking pose search. Following recent literature recommendations [84, 85], the top compounds from each run were re-scored for improved accuracy using `smina` [86] (see Figure 4).

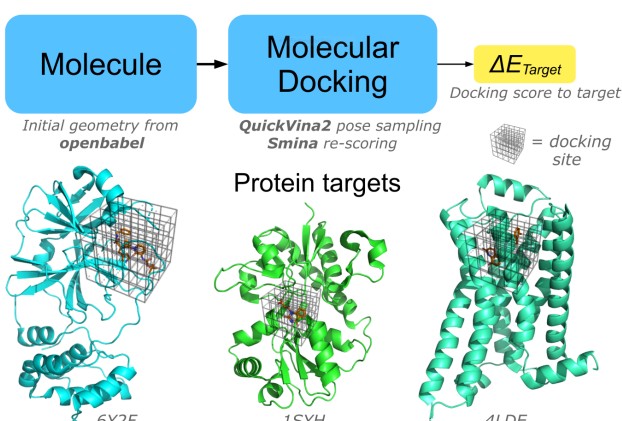

Figure 4: Overview of the computational workflow for protein ligands design benchmarks. The process accepts a SMILES string for protein targets 1SYH, 6Y2F, and 4LDE, generates Cartesian coordinates using `Open Babel`, and samples docking poses via `smina`. The lowest docking score is returned as the target property.

The results are compiled in Table 3, which includes the averages of the top docking scores achieved by each model. We also present the highest docking scores from the reference dataset for each of the tasks (`DATASET` in Table 3), along with the docking scores of the original ligands bound to the three target proteins in their respective crystal structures ("Native Ligand" in Table 3). Moreover, Table 3 indicates the percentage of sampled molecules that pass standard structure filters employed in drug discovery (see SR in Table 3). Notably, the native ligand for 6Y2F fails to pass these constraints, illustrating that these constraints only evaluate the likelihood of a structure being both stable and synthesizable. Across all protein targets, SMILES-VAE struggles to suggest structures with better

docking scores than in the reference dataset (-10.0/-10.4 for 1SYH, -8.3/-8.9 for 6Y2F, and -10.7/-11.1 for 4LDE), and it also yields the lowest success rates (12.3% for 1SYH, 13.1% for 6Y2F, and 12.6% for 4LDE). The SELFIES-VAE appears to perform slightly better for both docking scores (-10.4/-10.9 for 1SYH, -9.6/-10.1 for 6Y2F, and -11.1/-11.9 for 4LDE) and success rates (34.8% for 1SYH, 38.9% for 6Y2F, and 38.9% for 4LDE), but not significantly so. No single model consistently achieves the highest docking scores across all three targets. REINVENT attains the lowest average docking score for 1SYH (-11.8/-12.1), JANUS for 6Y2F (-11.3/-11.9), and GB-GA for 4LDE (-12.9/-13.8). Finally, due to the integration of our filters, the best designs in this set of benchmarks largely correspond to stable and potentially synthesizable compounds.

## 3.4 Design of Chemical Reaction Substrates

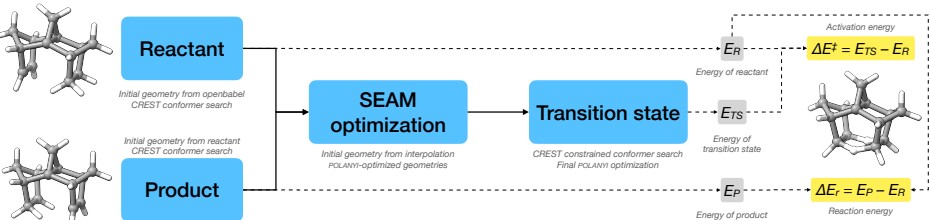

Figure 5: Overview of the workflow for designing chemical reaction substrates. Starting with a SMILES string, the code optimizes reactants and products, generates a guessed transition state via SEAM optimization, and conducts constrained conformational sampling. Reaction energy and approximate SEAM activation energy are then extracted.

Table 4: Results for chemical reaction substrate design benchmarks. Models, trained on a benchmark dataset generated with STONED-SELFIES cycles, yield mean and standard deviation of optimal objective values over five runs (mean $\pm$ standard deviation). Baselines include the best molecule in the training dataset ("Dataset") and the parent unsubstituted substrate ("Parent Substrate"). Metrics: $\Delta E^{\ddagger}$: Activation energy of the reaction; $\Delta E_r$: Reaction energy.

| | $\Delta E^{\ddagger}$ | $\Delta E_r$ | $\Delta E^{\ddagger} + \Delta E_r$ | $-\Delta E^{\ddagger} + \Delta E_r$ |
|---|---|---|---|---|
| PARENT SUBSTRATE | 85.16 | 0.00 | 85.16 | -85.16 |
| DATASET | 64.94 | -34.39 | 56.48 | -95.25 |
| SMILES-VAE | 76.81 $\pm$ 0.25 | -10.96 $\pm$ 0.71 | 71.01 $\pm$ 0.62 | -90.94 $\pm$ 1.04 |
| SELFIES-VAE | 72.45 $\pm$ 3.79 | -10.45 $\pm$ 3.83 | 72.05 $\pm$ 0.00 | -87.82 $\pm$ 2.13 |
| MoFlow | 70.12 $\pm$ 2.13 | -20.21 $\pm$ 4.13 | 63.21 $\pm$ 0.69 | -92.82 $\pm$ 3.06 |
| SMILES-LSTM-HC | 59.64 $\pm$ 4.10 | -31.03 $\pm$ 16.15 | 71.81 $\pm$ 1.56 | -91.58 $\pm$ 2.14 |
| SELFIES-LSTM-HC | 63.17 $\pm$ 4.34 | -21.02 $\pm$ 4.95 | 68.06 $\pm$ 5.74 | -96.59 $\pm$ 4.59 |
| REINVENT | 68.38 $\pm$ 2.00 | -24.35 $\pm$ 6.46 | 55.25 $\pm$ 5.88 | -94.52 $\pm$ 1.20 |
| GB-GA | 56.04 $\pm$ 3.07 | -41.39 $\pm$ 5.76 | 45.20 $\pm$ 6.78 | **-100.07 $\pm$ 1.35** |
| JANUS | **47.56 $\pm$ 2.19** | **-45.37 $\pm$ 7.90** | **39.22 $\pm$ 3.99** | -97.14 $\pm$ 1.13 |

Developing new chemical reactions and finding new catalysts for existing ones are important goals to drive innovations in drug and material discovery [87], and move towards more sustainable production [88]. Due to the importance of chemical reactions and the lack of reliable molecular design benchmarks related to reactivity, we looked for alternative approaches allowing us to model TSs in a reliable way with increased efficiency. We decided to use the SEAM force field method, which combines the force fields of two minima directly connected via the intrinsic reaction coordinate to the TS of interest generating an effective TS force field [89].

With this method, we chose to model the intramolecular concerted double hydrogen transfer reaction of *syn*-sesquinorbornenes, which is a dyotropic reaction of type II [90]. We use this reaction to define an inverse molecular design benchmark that targets modification of the substrate and product structures to alter reactivity. As main target properties defining reactivity in this system, we selected the corresponding activation and reaction energies. The first objective is to minimize the activation energy and corresponds to making the reaction faster, a common target when developing catalysts or reagents. Our second objective focuses on the thermodynamic driving force and aims to minimize the reaction energy making the respective product maximally thermodynamically favorable. Both the

third and the fourth task combine two properties into one objective function. The third corresponds to finding substrates that lead to both a fast and a thermodynamically favorable reaction. The fourth is about finding substrates that cause both a slow and a thermodynamically favorable reaction.

Like for the other benchmarks, we provide a reference dataset that can be used for training the generative models. As this benchmark requires molecules that contain the *syn*-sesquinorbornene motif, we needed to create the new SNB-60K dataset with approximately 60,000 molecules. The simulation workflow starts with a check of the required structural constraints (*vide supra*) in the proposed substrate SMILES string. If any constraint is violated, the workflow will be aborted and an extremely bad objective value of -10,000 returned. If all constraints are satisfied, initial guess Cartesian coordinates will be generated. Subsequently, a combination of conformer searches and constrained optimizations generates the reactant and product structures required to set up the SEAM force field and perform TS optimization. Finally, conformer search of the TS is carried out, followed by subsequent SEAM force field optimization, and the reaction and activation energies are derived from the lowest energy structures of reactant, TS and product. The workflow is shown in Figure 5.

The performance of the generative models considered on the molecular reactivity benchmarks is provided in Table 4. As similarly seen on the protein ligand design benchmarks, we observed that both the VAE models failed to surpass the best values in the dataset (e.g., SMILES-VAE: 76.81 versus 64.94 in the dataset for $\Delta E^{\ddagger}$, -10.96 versus -34.49 in the dataset for $\Delta E_r$, 71.01 versus 56.48 in the dataset for $\Delta E^{\ddagger} + \Delta E_r$, and -90.94 versus -95.25 in the dataset for $-\Delta E^{\ddagger} + \Delta E_r$). Compared to the VAEs, both the LSTM-HC models, MoFlow, and REINVENT perform better but they are not able to improve upon the best molecules from the dataset for all four benchmark objectives. In particular, SELFIES-LSTM-HC only reaches -21.02 versus -34.49 in the dataset for $\Delta E_r$ and 68.06 versus 56.48 in the dataset for $\Delta E^{\ddagger} + \Delta E_r$. Only JANUS and GB-GA consistently propose structures that outperform the best reference compounds. The best structures proposed in these benchmark tasks (cf. Supporting Information) generally show that the majority of structures is reasonable in terms of stability. In addition, the inclusion of SAscore constraints in the two combined objectives seems to particularly result in smaller molecules as molecular size tends to correlate with the SAscore.

## 3.5   Model Timing Comparison

Finally, one overlooked consideration for choosing generative models is the computation time needed for pre-conditioning the model based on a reference dataset and for proposing new candidates. The main reason we conducted this benchmark is our belief that shorter pre-conditioning and structure generation translates to increased usage of molecular design algorithms. While it is a minor overhead in the case of time-limiting property evaluation, it can constitute a significant entrance barrier for testing a model in benchmark tasks with comparably fast evaluations before using it on the actual problem of interest. We used three different metrics for comparison based on the reference dataset of the design of protein ligands. First, we evaluated the time the models need for pre-conditioning (or training) when provided with the corresponding reference dataset. For that, the models have access to 24 CPU cores and 1 GPU. Second, as access to GPUs can sometimes be limited, we wanted to evaluate the impact of GPU usage on the model pre-conditioning time. However, for some of the models tested, training times were too prohibitive for benchmarking when no GPU was provided. Thus, to measure the impact of using a GPU for training, instead of the full time, we evaluated single epoch times both with and without access to a GPU. Third, we determined the time required by the algorithms to propose 10,000 unique structures. All timing results are illustrated in Figure 6.

Overall, we find that both VAEs needed by far the longest training time, both taking considerably longer than 9 hours, with the SELFIES-VAE requiring somewhat more time than the SMILES-VAE. In contrast, both LSTM-HC models only need approximately 2 hours with no statistically significant difference between using SMILES and SELFIES. Compared to the other deep generative models, both MoFlow and REINVENT show by far the fastest training times finishing already after less than 1 hour in our benchmark. When looking at the single epoch times with and without GPU usage, we find that the relative order with respect to timing is comparable regardless of whether a GPU is utilized. It is particularly notable that both the VAEs have single epoch times of approximately 8 hours which leads to an estimated training time without a GPU of about 25 days, which we decided to be too prohibitive to perform full training. All other deep generative models have single epoch times significantly faster than 1 hour leading to estimated training times below 1 day. The fastest models in that respect are again MoFlow and REINVENT with single epoch times of only several

minutes. Notably, MoFlow has a faster epoch time than REINVENT with GPU usage, but a slower one without. Nevertheless, the total pre-conditioning times of both GAs, which are provided in the diagram depicting single epoch times (cf. Figure 6C), is still shorter which only surmounts to several seconds without the use of a GPU.

## 4   Conclusion and Outlook

We developed TARTARUS, a modular set of realistic molecular design objectives representative of common problems in the domains of organic materials, drugs, and chemical reactions to be used as general goal-directed benchmarks for generative models. Most tasks are directly inspired by design objectives pursued in the literature. Hence, high-performing molecules have potential value in real applications. We observed that none of the representative generative models selected consistently outperformed the others across all the design tasks. This is due to a seemingly strong dependence of the performance of some models on the specific benchmark domain and suggests the importance of conducting diverse molecular design benchmarks that reflect actual problems pursued in chemistry to evaluate the generalizability of

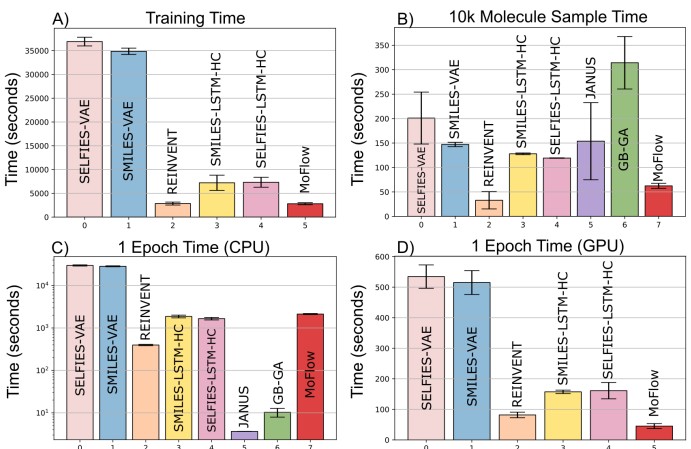

Figure 6: Results of model timing comparisons. Models are trained on a subset of the DTP Open Compound Collection. The benchmark metrics consist of model training time (or precondition time) with the use of a GPU (A), single epoch times during training both with (D) and without a GPU (C), and the time required to sample 10,000 structures (B). Results are provided as mean (main bar) and standard deviation (error bars) of the metrics obtained in five independent runs.

inverse molecular design algorithms across domains. Furthermore, it shows that current molecular design algorithms have significant room for improvement in terms of generality, particularly deep generative models. Thus, more sophisticated approaches do not guarantee stronger inverse design performance , especially when not tested on the problem domain of interest.

We note that relatively simple approaches like genetic algorithms that do not need large reference datasets show more consistently good performance across benchmarks, whereas the VAE approaches, that strongly rely on the available training data, show limited performance across multiple benchmarks. However, more comprehensive investigations are required to confirm this hypothesis and identify the origins of the limited performance of some of the approaches tested. Moreover, we believe that demonstrating strong performance of computer algorithms applied to realistic design objectives will inspire researchers across fields to adopt these techniques in their workflows eventually leading to mainstream use. Accordingly, we believe that TARTARUS will act as a stepping stone for advancing deep generative models and for inspiring the development of realistic molecular design benchmarks. Notably, we would like to emphasize that TARTARUS does not replace the development of more specialized molecular design benchmarks that are tailored to specific domains of interest and necessarily follow a different design workflow, particularly in drug design [91–94]. We believe that such specialized benchmarks are complementary to TARTARUS and are equally important to assess the practicality and performance of molecular generative models.

Nevertheless, this first work on TARTARUS still leaves room for further developments. As is common in real-world molecular design, objectives need to be revised when undesired structures are promoted and new property requirements can emerge. In particular, the relative importance of multiple target properties sometimes needs to be adjusted and the consideration of additional properties becomes necessary. Furthermore, the benchmark domains currently covered are far from comprehensive,

providing opportunities for the development of additional design tasks in the near future. Deep generative models capable to design 3D molecular structures directly are currently not well supported, as proposed conformers would be ignored. Extending TARTARUS in that regard requires including geometries in our reference datasets. Both for the protein ligand and the chemical reaction substrate design benchmarks, specialized geometries are needed. Hence, it would not be enough to train 3D generative models suggesting ground-state gas-phase geometries. The chemical reaction substrate design benchmarks require multiple geometries as both two ground state geometries and one transition state geometry are simulated. Consequently, most current 3D generative models cannot fulfill this requirement out of the box and could thus not replace established global geometry optimization strategies. Finally, the results of 3D generative models require dedicated performance comparisons as molecular properties are intrinsically tied to their 3D structures. For a meaningful comparison to the other molecular design algorithms considered, it is necessary to separate 3D structure generation from molecular connectivity generation. This requires introducing separate conformer generation benchmarks for each of the benchmark tasks. Accordingly, these extensions of TARTARUS will be the subject of future work. Finally, we aim to launch open molecular design challenges on a regular basis making use of the benchmark codes included in TARTARUS. This will allow us to provide regular measurements of the state of the art in molecular generative models and promote further advances in the field.

## Data Availability

Code and datasets for running all the TARTARUS benchmarks are provided in our public GitHub repository: https://github.com/aspuru-guzik-group/Tartarus (DOI: https://zenodo.org/badge/latestdoi/444879123). For further discussions and collaboration, we invite readers to become part of our Discord community: https://discord.gg/KypwPXTY2s.

## Acknowledgements

AK.N. acknowledges funding from the Bio-X Stanford Interdisciplinary Graduate Fellowship (SGIF). R.P. acknowledges funding through a Postdoc.Mobility fellowship by the Swiss National Science Foundation (SNSF, Project No. 191127). G.T. acknowledges the Natural Sciences and Engineering Research Council of Canada for support through the Postgraduate Scholarships-Doctoral (PGS-D) program. K.J. acknowledges funding through an International Postdoc grant from the Swedish Research Council (No. 2020-00314). A.A.-G. thanks Anders G. Frøseth for his generous support. A.A.-G. acknowledges the generous support of Natural Resources Canada and the Canada 150 Research Chairs program. Some computations were performed on the Béluga and Narval supercomputers situated at the École de technologie supérieure in Montreal. We also thank the SciNet HPC Consortium for support regarding the use of the Niagara supercomputer. SciNet is funded by the Canada Foundation for Innovation, the Government of Ontario, Ontario Research Fund - Research Excellence, and the University of Toronto. Computations were also performed on the Cedar supercomputer situated at the Simon Fraser University in Burnaby. In addition, we acknowledge support provided by Compute Ontario and Compute Canada. Some of the computing for this project was also performed on the Sherlock cluster. We would like to thank Stanford University and the Stanford Research Computing Center for providing computational resources and support that contributed to these research results. We thank the U.S. Department of Energy (DOE)/NREL/ALLIANCE for making the Reference Air Mass 1.5 Spectra (AM1.5G) publicly available at https://www.nrel.gov/grid/solar-resource/spectra-am1.5.html.

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

# Supplementary Information:

# TARTARUS: Practical and Realistic Benchmarks for Inverse Molecular Design

## S1 Introduction

Traditionally, property-guided optimization has relied on expert intuition [95] and several rounds of trial, error, and human-inspired optimization, occasionally supported by computer simulations. Alternatively, computer-assisted approaches have employed virtual screening [96] or optimization algorithms such as genetic algorithms (GAs) [97, 98, 5]. More recently, with the surge of deep learning, deep generative models have emerged, specifically designed to operate in chemical space and tackle inverse molecular design [99, 100, 1]. This has led to the development of numerous algorithmic approaches for this purpose, with the most popular including variational autoencoders (VAEs) [101, 31], generative adversarial networks (GANs) [102, 103], and reinforcement learning (RL) [104, 12].

## S2 Methods Overview

In this section, we provide an overview of the molecular generative models employed throughout this work and summarize the associated design choices we needed to make during their implementation. The molecular design algorithms we considered are VAEs, long short-term memory hill climbing (LSTM-HC) models [105, 33, 3], REINVENT [30], JANUS [7], and a graph-based genetic algorithm (GB-GA) [34]. At the core of the majority of these approaches are molecular string representations, the most commonly used of which is the Simplified Molecular Input Line Entry System (SMILES) [46]. Accordingly, many of the algorithms tested rely on predicting subsequent characters from partial strings to propose structures. However, algorithms based on SMILES can regularly produce invalid strings that do not represent molecules, which is problematic both in terms of robustness and interpretability of the corresponding methodologies [19, 20]. Consequently, this issue was addressed systematically by introducing Self-Referencing Embedded Strings (SELFIES) [19], a molecular string representation that guarantees validity. Thus, unlike for SMILES, every arbitrary combination of SELFIES characters represents a molecule. Nevertheless, its impact on structure optimization has not yet been evaluated systematically [20]. To address this issue, we modify some of the existing generative models relying on SMILES to be also compatible with SELFIES and test their performance depending on representation, similar to how it has been done recently [18].

Among the models tested, REINVENT, the VAEs, and the LSTM-HC models use recurrent neural networks (RNNs) [106] to learn the conditional probability distributions of the characters that represent molecules. RNNs are a class of artificial neural networks (ANNs) that utilize sequential information from their previous predictions and states. They have been incorporated into molecular design algorithms with special NN node architectures such as gated recurrent units (GRUs) [107] and LSTM cells [105]. The first of these models, i.e., REINVENT [30], is an RL-based approach that relies on an LSTM-based RNN as an agent which is tasked with generating molecules with desired properties. Over time and continued training, the agent learns to propose compounds with increasingly favorable target property values.

For the VAEs, we used the implementation described by Gómez-Bombarelli et al. as a starting point [31]. Therein, SMILES are converted to one-hot encodings and, subsequently, passed through a 1-dimensional convolutional neural network (CNN) encoder that generates a continuous latent space. A separate RNN decoder with GRU nodes regenerates the SMILES strings from the latent space. Molecular optimization is performed on the continuous representation of the latent space in a stochastic and iterative manner. Specifically, the best available molecule for a given task is encoded into the latent space of a trained VAE. Subsequently, random Gaussian noise is added to the obtained latent vector to produce a population of latent vectors that can be decoded to a population of new candidate structures. These structures are evaluated and the best compound obtained is used as a seed in the subsequent iteration. In our experience, this strategy leads to more stable optimization compared to direct gradient-based optimization in the latent space as described in the literature [31]. Importantly, we modified the original implementation to be compatible with both SMILES

and SELFIES (cf. Computational Details in the Supporting Information). These variations will be referred to as SMILES-VAE and SELFIES-VAE, respectively.

As their name implies, the LSTM-HC models [105, 33, 3, 108] rely on LSTM cells in the NN architecture to model the character sequence probability distributions. After initial training, the resulting model is sampled using randomly truncated strings of the best currently available molecules for a given task. These truncated strings are used as seeds and completed stochastically by sampling the learned conditional probability distributions to produce a population of candidate structures. The best new compounds obtained after property evaluation are used to retrain the model and initiate a subsequent sampling cycle. Thus, iterative sampling and retraining gradually improves the designs of the LSTM-HC models. Again, we test both the original implementation of LSTM-HC that relies on SMILES [33, 3], which will be referred to as SMILES-LSTM-HC, and a modified version making use of SELFIES, referred to as SELFIES-LSTM-HC. Notably, the SMILES-LSTM-HC model was among the best performing models in the original publication of the GuacaMol benchmarks [3] (referred to as "SMILES LSTM" in that reference).

Before the increasing adoption of ML approaches for molecular design, GAs [109, 5, 110, 111] were among the most popular computer-based methods for that purpose. Inspired by natural selection as defined in Darwinian evolution, GAs are heuristic population-based optimization algorithms. They rely on repeated stochastic generation of offspring from an existing population of candidate solutions via the genetic operations mutation and crossover. Subsequently, selection of the candidate solutions with the highest fitness determines the population to be propagated to the next generation, starting another iteration. In this work, we consider two specific implementations for inverse molecular design, GB-GA [34], and JANUS [7]. The former leverages structural information from a dataset of reference molecules by mimicking the corresponding distribution of atoms and bonds when performing genetic operations. Notably, in the original publication of the GuacaMol benchmarks [3], GB-GA achieved the best overall performance. In contrast, JANUS is a GA augmented by an NN classifier [7] (referred to as "JANUS+C" in the original publication) that actively judges the quality of a molecule before performing a full fitness evaluation. Molecules classified as likely possessing high fitness are then propagated to the next generation for subsequent evaluation. For mutation and crossover, JANUS relies on STONED-SELFIES [112], a set of algorithms based on the guaranteed validity of SELFIES to perform structural modifications. Additionally, unlike GB-GA, JANUS maintains two distinct molecule populations, one explorative and one exploitative with respect to conducted structural changes [7]. These populations exchange some members at every iteration but otherwise explore the chemical space independently [7].

When using TARTARUS, the following procedures should be adopted to obtain benchmark results that are consistent with the ones provided herein. The first step for running one of the benchmarks, if necessary, is to train the generative model on the provided dataset. For all the ML models, we used the first 80% of the reference molecules for training and the remaining 20% for hyperparameter optimization. Then, the (trained) model is tasked with proposing structures to be evaluated by the objective function of the corresponding benchmark task. Notably, structure optimization was always initiated using the best reference molecule from the corresponding dataset. For the benchmarks concerned with designing photovoltaics, organic emitters, and protein ligands, structure optimization was carried out with a population size of 500 and a limit of 10 iterations, leading to a maximum number of 5,000 proposed compounds overall. For the design of chemical reaction substrates, we used the same maximum number of proposed compounds but used a population size of 100 and limited the number of iterations to 50 instead. Additionally, the associated run time was limited to 24 hours, which resulted in termination for several molecular design runs before reaching 5,000 molecule evaluations. Furthermore, to increase robustness and reproducibility of our results, we repeated each optimization run five times, allowing us to report the corresponding outcomes with both an average and a standard deviation. We believe that this resource-constrained comparison approach is necessary for fairly comparing methods and should be used as a standard by the community. A detailed account of the parameters and settings used for running each of the models is provided in the Computational Details section of the Supporting Information.

# S3   Additional Results

## S3.1   Design of Organic Photovoltaics

OPVs offer simplified, cost-effective production [113, 35] and enhanced mechanical properties, particularly regarding specific weight and flexibility [38, 39]. Despite significant progress, OPVs still have lower power conversion efficiencies (PCEs) and shorter device lifetimes [114], prompting ongoing research efforts in molecular design for OSCs [115, 114, 39].

The simplest OPV cells comprise two electrodes and a photoactive layer for photoconversion, typically containing two distinct materials: donor and acceptor [35]. In bulk heterojunction cells [116], donors and acceptors are mixed, allowing nanoscale phase separation to maximize interfacial area and minimize distance within the phases [117]. Upon light exposure, excitons—neutral quasi-particles consisting of bound electron-hole pairs [118, 117, 35]—form in the photoactive layer with limited lifetimes and diffusion lengths [117]. The bulk heterojunction architecture enables excitons to reach the interface, dissociating into a free electron and hole that become charge carriers across the phases [118, 117, 35]. These charge carriers travel through the photoactive layer to the electrodes, generating a photocurrent [35]. For exciton charge separation at the donor-acceptor interface to occur, the process must be energetically neutral or favorable [118, 35]. Therefore, the exciton energy, estimated by the HOMO-LUMO gap of the light-absorbing material, must be greater than or equal to the effective heterojunction bandgap [118, 35], approximated by the energy difference between the donor's highest occupied molecular orbital (HOMO) and the lowest unoccupied molecular orbital of the acceptor (LUMO) [41]. Notably, PCE is the percentage of power from the incident solar irradiation that is converted into electricity by the OPV device. We believe that the reduced reference dataset presents a more realistic scenario for new molecular design projects and is better suited for benchmarking generative models, especially when property simulations are time-consuming. We propose the following two molecular design benchmark objectives:

1. Maximize the following function:
   $PCE_{PCBM} - SAscore.$

2. Maximize the following function:
   $PCE_{PCDTBT} - SAscore.$

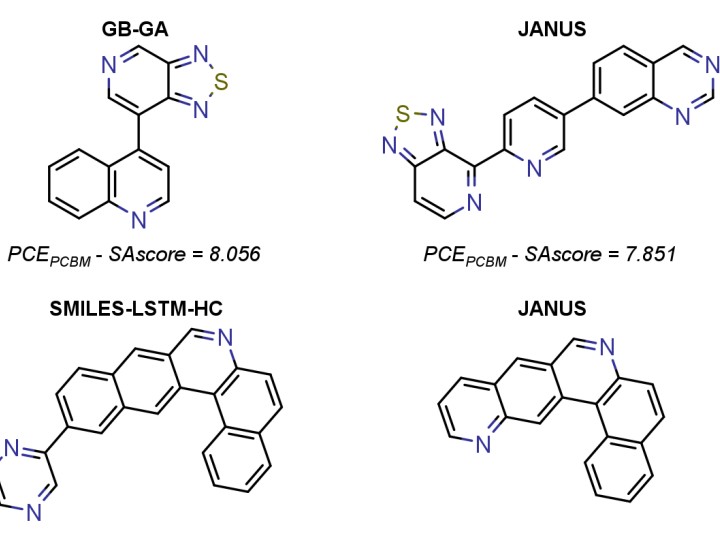

Figure S1: Best molecules found in each of the benchmark tasks inspired by the design of organic photovoltaics. Additionally, the corresponding objective values and the molecular design models that proposed the structures are indicated.

The best molecules found in each of the design of organic photovoltaics benchmark tasks together with the corresponding objective values and the model that proposed them are depicted in Figure S1.

## S3.2 Design of Organic Emitters

The next set of benchmarks is inspired by the design of purely organic emissive materials for organic light-emitting diodes (OLEDs), which received significant attention in recent years [55–57] after the discovery of thermally activated delayed fluorescence (TADF) in the field [60]. Their main applications are digital screens and lighting devices [55]. For the former application, compared to alternative technologies, OLEDs offer improved image quality and enable both lighter and thinner devices [55]. For the latter, OLEDs are potentially more energy-efficient [55]. In OLED devices relying on TADF, after electric excitation, light generation takes place from the first excited singlet state via fluorescence [60]. However, only 25 % of the initial excitons are excited via an electric current into singlet states and contribute directly to light emission via excited state thermal relaxation and subsequent prompt fluorescence [55–57, 60]. The 75 % of the initial excitons that are excited into triplet states relax thermally to the corresponding first excited triplet states [55–57, 60]. However, in ordinary organic molecules, light emission from their first excited triplet states via phosphorescence is slow, giving rise to various radiationless decay and decomposition pathways [55–57]. Consequently, both device efficiency and device lifetime are reduced considerably unless these triplet excitons can still be utilized for light production [55–57]. To achieve that, TADF emitters rely on minimizing the energy difference between the first excited singlet and triplet states, i.e., the singlet-triplet gap, allowing thermal upconversion of the triplet excitons to the first excited singlet state giving rise to delayed fluorescence [55–57]. Importantly, under the assumption of fast both forward and reverse intersystem crossing (ISC), the steady-state triplet population is governed by the singlet-triplet gap and its reduction leads to reduced triplet population and acceleration of delayed fluorescence. This increases internal quantum efficiency up to a maximum of 100 % and reduces decomposition of the emissive material [55–57]. Designing efficient emissive organic materials for blue OLEDs is of particular interest. This can be achieved by targeting organic molecules with excitation energies between their ground state and their first excited singlet states that correspond to the energy of blue light. The fitness functions of these three tasks are summarized as follows:

- Minimize the singlet-triplet gap, $\Delta E(S_1\text{-}T_1)$.
- Maximize the oscillator strength for the transition between $S_1$ and $S_0$, $f_{12}$.
- Maximize the following function:
  $+f_{12} - \Delta E(S_1\text{-}T_1) - |\Delta E(S_0\text{-}S_1) - 3.2 \ eV|$.

The best molecules found in each of the design of organic emitters benchmark tasks together with the corresponding objective values and the model that proposed them are depicted in Figure S2.

## S3.3 Design of Protein Ligands

Notably, while docking simulations are a standard tool in virtual screening pipelines of drug discovery campaigns, their accuracy compared to experimental binding affinities is at best modest [119–122]. Thus, typical workflows use them merely for a preselection which is narrowed down further with subsequent free energy simulations [123]. Nevertheless, for molecular design benchmarks, docking still provides the best trade-off between computational efficiency and relevance to real-world molecular design. Additionally, it is important to realize that drug design requires the consideration of many other molecular design aspects that make or break a hit compound, e.g., toxicity, solubility, stability and many more. However, as these properties are significantly harder to model computationally, we decided to disregard them for our set of benchmarks. Notably, finding small molecule ligands for the selected proteins marks the first step towards the development of treatments for various important conditions.

Importantly, most likely due to its maturity, complexity, and demand for resources, drug design was probably the first chemical problem where molecular design algorithms were tested comprehensively.

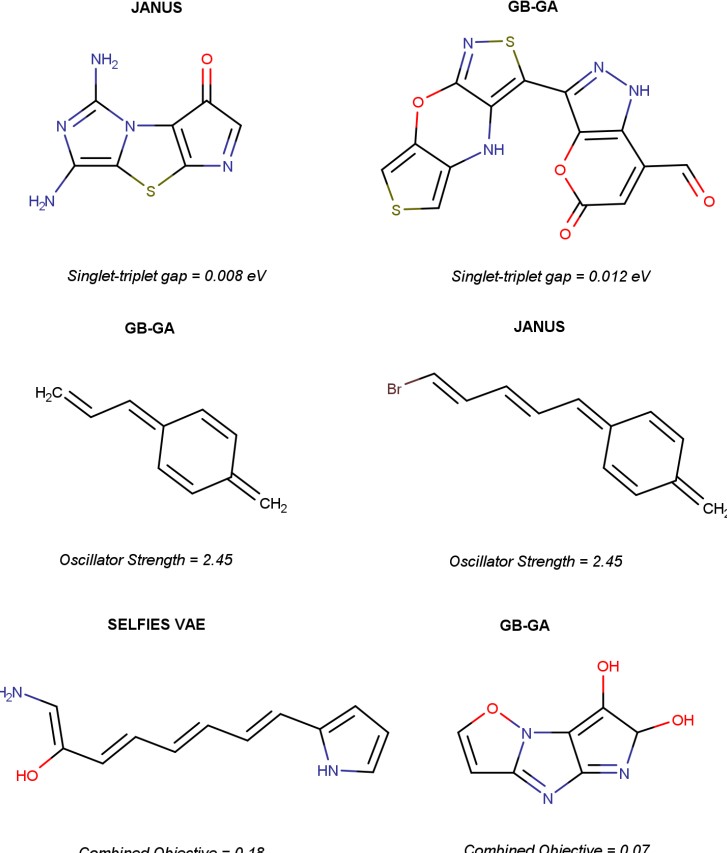

Figure S2: Best molecules found in each of the benchmark tasks inspired by the design of organic emitters. Additionally, the corresponding objective values and the molecular design models that proposed the structures are indicated.

The use of computer algorithms has a long-standing history in medicinal chemistry and GA-based molecular design algorithms making use of full atomic representations have already been used as early as 1993 [6]. Initial toy tasks for testing these algorithms included rediscovery of known drugs via a structural similarity metric, highly simplified molecular docking of ligands to rigid protein binding sites minimizing the interaction scores, and the optimization of estimated molecular properties like the decadic logarithm of the *n*-octanol-water distribution coefficient (log P) [6]. Interestingly, some of the still most widely used benchmarks for generative models rely on essentially the same types of metrics as they are simple to implement and compute [2, 14, 3, 124, 29]. More recently, the quantitative estimate of drug-likeness (QED) was introduced, which is a desirability function that uses the position of a small set of common and simple molecular descriptors relative to the corresponding distributions in a dataset of approved drugs to estimate structural resemblance to therapeutics [125]. Due to its simplicity, it found immediate application in several molecular design benchmarks [126, 103, 127]. However, using QED alone in generative models is not meaningful for finding drug candidates as it only accounts for the general structural requirements of drug-like molecules but disregards intended modes of action with respect to specific targets entirely. The specific benchmark objectives we implemented are summarized below.

- Minimize the docking score to 1SYH, $\Delta E_{1SYH}$.

- Minimize the docking score to 6Y2F, $\Delta E_{6Y2F}$.

- Minimize the docking score to 4LDE, $\Delta E_{4LDE}$.

Notably, the corresponding objective functions do not solely consist of the docking scores but also have hard structural constraints directly incorporated. When these constraints are not fulfilled, an extremely unfavorable score of 10,000 is returned instead of the actual docking score. They consist of a set of filters that checks for the presence of unstable or reactive structural moieties and determines whether the compound in question fulfils Lipinski's Rule of Five [128]. Notably, most of these filters were developed based on our previous experience using molecular design algorithms to minimize docking scores and are tailored to avoid extremely unstable and reactive molecules that seem to be strongly favored by molecular docking simulations [7]. Additionally, the constraints avoid rings with more than 8 members as the docking approach implemented is unable to sample the corresponding conformations in a proper manner [129]. To fulfill them, the proposed structure needs to have an SAscore smaller than 4.5, which is the revised optimal threshold for that metric proposed in the literature [130], and a QED value larger than 0.3, which corresponds to the first quartile of the distribution of QED values for compounds in the CHEMBL database [125]. A list of these metrics is provided here:

- Absence of reactive groups.
- Absence of formal charges.
- Absence of radicals.
- At most 2 bridgehead atoms.
- No rings larger than 8-membered.
- Fulfils Lipinski's Rule of Five.
- $SAscore < 4.5$.
- $QED > 0.3$.
- $TPSA > 140$.
- Molecule passes the PAINS and WEHI and MCF filters.
- Molecule does not contain Si and Sn atoms.

The best molecules found in each of the design of protein ligands benchmark tasks together with the corresponding objective values and the model that proposed them are depicted in Figure S3.

## S3.4   Design of Chemical Reaction Substrates

Whereas, classically, the optimization of reaction parameters was largely dominated by experimental work, in recent years, the significant increase in computing power and the continuous improvement of computer algorithms enabled molecular simulations to play an increasingly important part [131–134]. With the aid of transition state (TS) theory, fundamental reaction parameters such as thermodynamic feasibility, reaction rate and selectivity can be computed from first principles [135, 136]. This requires explicit modeling of the corresponding TS, a postulated state on the multi-dimensional potential energy surface (PES) of the process, which lies on a saddle point of order one [137]. Due to the difficulty of finding such saddle points in high-dimensional spaces and the often delicate electronic structure associated with the corresponding structures, in practice, automated TS optimizations often suffer from high failure rates in the range of 10–50% [138–140], and even well-behaved case studies combined with robust methodologies still have some room for improvement in that respect[141]. Additionally, they are typically carried out with relatively resource-intensive density functional approximation (DFA) calculations taking on the order of hours or even days to complete [138, 140]. Overall, these issues make them ill-suited for benchmarking molecular design algorithms or for routine application combined with generative models in computer-guided inverse molecular design campaigns.  Nevertheless, GAs have been employed for computational catalyst design, particularly via the use of regression models based electronic structure descriptors derived from SQC simulations [142] and based on both structural and electronic structure descriptors obtained from DFA computations [143].  Most notably, very recently, GAs have been employed to optimize an organocatalyst for the Morita-Baylis-Hilman reaction [144].

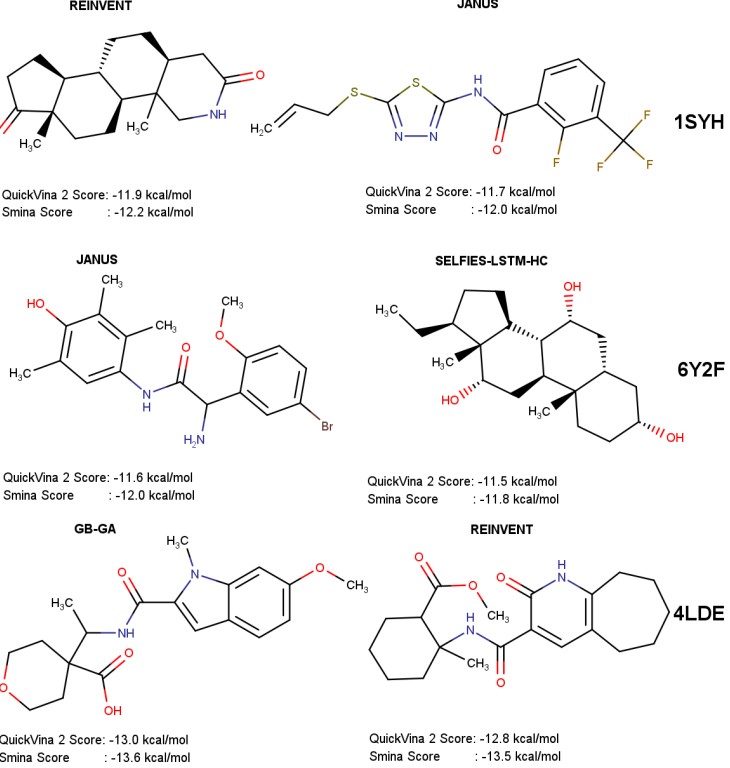

Figure S3: Best molecules found in each of the benchmark tasks inspired by the design of protein ligands. Additionally, the corresponding objective values and the molecular design models that proposed the structures are indicated.

The two force fields cross at approximately the TS geometry, and we introduce a coupling term that turns this into an avoided crossing, ensuring smoothness of the PES [89]. The resulting PES has a local maximum on the ground state surface (the TS) and a corresponding local minimum on the excited state surface, at approximately the same geometry. This allows optimizing TS geometries with robust gradient-based minimization algorithms. Thus, the SEAM force field method delivers activation energy estimates for reasonably sized molecules within minutes, and, in our hands, reaches very high success rates above 99.9% on a set of test reactions. Notably, we implemented this method in our Python package `polanyi`, which will be described in more detail in a separate publication. To generate the reference dataset for this set of benchmarks, starting from the unsubstituted reactive core structure, we performed repetitive cycles of STONED-SELFIES mutations [112] followed by removing all proposed compounds that violated the core and functional group constraints (Details in the Supporting Information). Thus, after several cycles, we obtained approximately 60,000 molecules defining the reference structures, which we refer to as SNB-60K dataset. Notably, in the selected reaction, there is only one TS connecting reactants and products in the selected reaction. Additionally, the fourth task aims to break the Bell–Evans–Polanyi principle [145–148], a linear free energy relationship that holds empirically for a large number of reactions. The following list summarizes the four benchmark tasks for chemical reactivity.

- Minimize the activation energy, $\Delta E^{\ddagger}$, and maintain the core and functional group constraints.

- Minimize the reaction energy, $\Delta E_r$, and maintain the core and functional group constraints.

- Minimize the following function:
  $+\Delta E^{\ddagger} + \Delta E_r$, and maintain the core, functional group and SAscore constraints.

- Minimize the following function:
  $-\Delta E^{\ddagger} + \Delta E_r$, and maintain the core, functional group and SAscore constraints.

On top of these primary objectives, we added several hard constraints that the target molecules need to fulfill in order to reward generative models that propose realistic and feasible molecules. In particular, all substrates need to retain the *syn*-sesquinorbornene motif (referred to as "core constraint") which is required for the reaction to take place. Additionally, we selected a set of unstable and reactive substructures that need to be avoided (referred to as "functional group constraint", details in section S5.2.4 of the Supporting Information). Furthermore, for the two benchmark tasks with objective functions combining two target properties, we also required all proposed structures to possess an SAscore [44] of 6.0 or lower (referred to as "SAscore constraint"). The following list summarizes the four benchmark tasks for chemical reactivity.

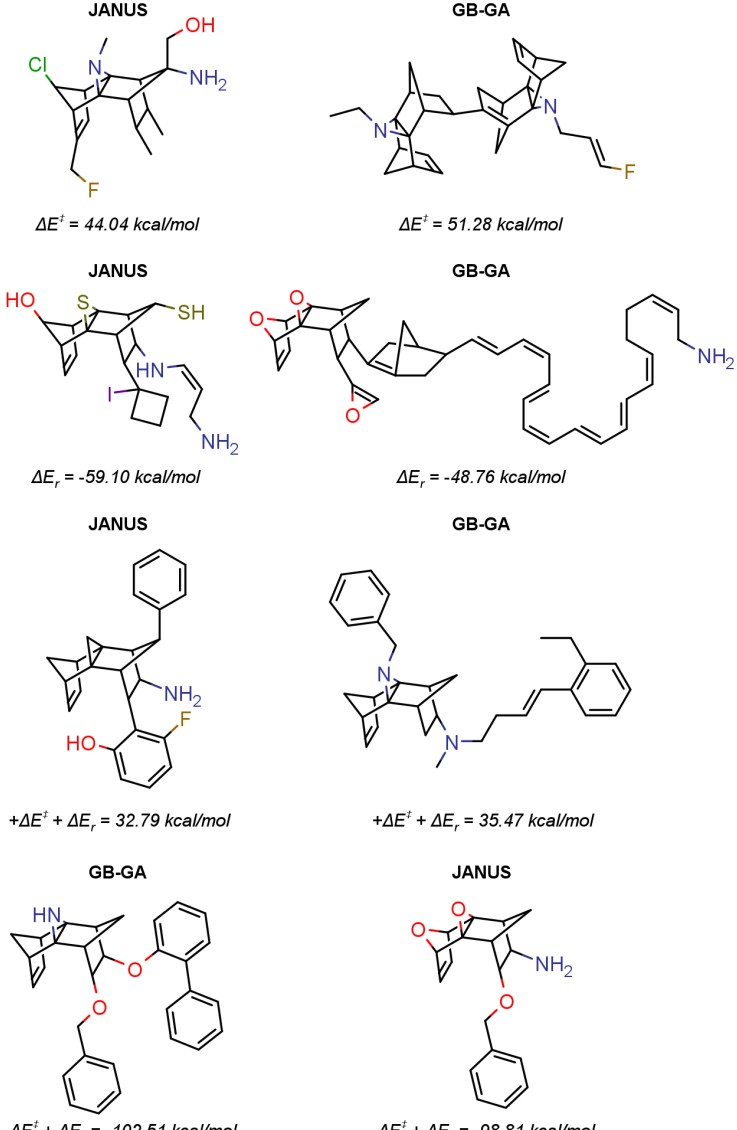

Figure S4: Best molecules found in each of the benchmark tasks inspired by the design of chemical reaction substrates. Additionally, the corresponding objective values and the molecular design models that proposed the structures are indicated.

The best molecules found in each of the design of chemical reaction substrates benchmark tasks together with the corresponding objective values and the model that proposed them are depicted in Figure S3.

## S3.5 Model Timing Comparison

For ML-based molecular design algorithms, the pre-conditioning corresponds to training time and sampling time, respectively. For the GA-based algorithms considered, they translate to the duration of the derivation of genetic operators from a reference dataset, and of applying the genetic operators to propose new candidate solutions, respectively. When the number of property evaluations is kept constant, assuming that the molecular size distribution between the generative models does not differ significantly, these steps are the major origin of timing differences between the algorithms. Notably, as was done for generating the molecular design results, we derived these timing metrics from five independent measurements and provide the results as averages with standard deviations.

Comparison of the single epoch times with and without a GPU allows estimating which models profit most from GPU usage. We find that the longer the single epoch time, the more the model profits from using a GPU which is consistent with expectations. Finally, looking at the sampling times, we find that REINVENT significantly outperforms all other methods considered needing less than 1 minute. Most of the other molecular design algorithms need between 2 and 3 minutes with GB-GA having the slowest sampling time of 6 minutes. Nevertheless, the differences between the methods are less pronounced here.

Table S5: Raw values for the timing benchmarks. Mean and standard deviation (mean $\pm$ s.d) timings for different models are provided based on five independent runs. N.A. means not applicable.

| | TRAINING TIME [S] | EPOCHS | SAMPLE TIME [S] | CPU EPOCH TIME [S] | GPU EPOCH TIME [S] |
|---|---|---|---|---|---|
| SELFIES-VAE | $36910 \pm 912$ | $75.6 \pm 5.9$ | $201 \pm 53$ | $29810 \pm 949$ | $535 \pm 38$ |
| SMILES-VAE | $34868 \pm 667$ | $74.1 \pm 6.2$ | $154 \pm 79$ | $28476 \pm 724$ | $515 \pm 39$ |
| MOFLOW | $2804 \pm 216$ | $70.1 \pm 5.4$ | $62.41 \pm 5$ | $2131 \pm 63$ | $45 \pm 8$ |
| GB-GA | N.A. | N.A. | $314 \pm 54$ | $3.653 \pm 0.007$ | N.A. |
| JANUS | N.A. | N.A. | $147 \pm 4$ | $10.3 \pm 2.4$ | N.A. |
| REINVENT | $2844 \pm 310$ | $33.8 \pm 1.2$ | $33 \pm 18$ | $397 \pm 14$ | $81.7 \pm 9.2$ |
| SMILES-LSTM-HC | $7208 \pm 1605$ | $45.8 \pm 10.5$ | $128.0 \pm 1.7$ | $1870 \pm 139$ | $157.4 \pm 5.6$ |
| SELFIES-LSTM-HC | $7321 \pm 1039$ | $45.4 \pm 6.3$ | $119.3 \pm 0.5$ | $1661 \pm 112$ | $161 \pm 27$ |

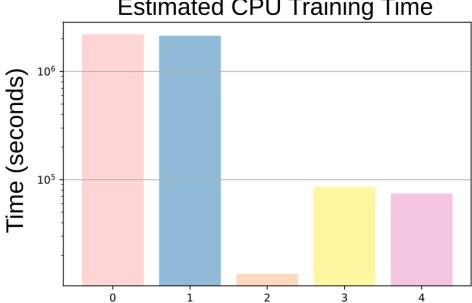

Figure S5: Estimated CPU training time for the deep generative models based on the CPU single epoch time and the number of epochs during training with GPUs. Models are trained on a subset of the DTP Open Compound Collection. Results are provided as mean (main bar) from five independent runs. The numbers on the abscissa each refer to the following molecular design algorithms. Model 0: SELFIES-VAE; Model 1: SMILES-VAE; Model 2: REINVENT; Model 3: SMILES-LSTM-HC, Model 4: SELFIES-LSTM-HC.

The numerical results of all the timing benchmarks are provided in Table S5. The results of the estimation of total CPU training times are illustrated in Figure S5.

## S3.6 Diversity Calculation

Following the definition of diversity in the literature[149], for each task, we calculate diversity of the proposed moleucles using the following equation:

$$\text{Diversity} = 1 - \frac{2}{n(n-1)} \sum_{X,Y} \text{sim}(X,Y) \tag{1}$$

The expression $\text{sim}(X,Y)$ computes the pairwise molecular similarity for all $n$ structures calculated as the Tanimoto distance of the Morgan fingerprints, which were obtained with a radius of size 3 and a 2048 bit size [150]. The individual results for the diversity evaluations of all the benchmark tasks are provided in Tables S6-S9. Overall, we observe mostly relatively small differences in the diversity of the proposed molecules between the different models, except for the chemical reaction substrate design benchmarks. We hypothesize that the strict structural requirement of the corresponding tasks are responsible for this observation as only molecules with the correct reactive core structure are subjected to the property simulation workflow. Additionally, we observe that all generative models except for SMILES-LSTM-HC show consistently a relatively high diversity across all four groups of benchmarks. The molecules proposed by SMILES-LSTM-HC for the organic photovoltaics and for the chemical reaction substrate benchmarks have quite a low diversity compared to the other generative models. Nevertheless, these results suggest that the diversity metric can be insightful in cases of especially low values, but otherwise does not allow to distinguish between the performance of the generative models in a reliable way.

Table S6: Diversity results for the organic photovoltaics benchmarks. Models are trained on a subset of the Harvard Clean Energy Project Database. Results are provided as mean and standard deviation of the best target objective values that are obtained in five independent runs in the form mean $\pm$ standard deviation. Metrics: $PCE_{PCBM}$: PCBM power conversion efficiency; $PCE_{PCDTBT}$: PCDTBT power conversion efficiency; SAscore: synthetic accessibility score.

|  | $PCE_{PCBM} - SAscore$ | $PCE_{PCDTBT} - SAscore$ |
|---|---|---|
| SMILES-VAE | $72.4 \pm 2.6\%$ | $81.6 \pm 3.0\%$ |
| SELFIES-VAE | $\mathbf{91.1 \pm 0.3\%}$ | $\mathbf{91.6 \pm 0.3\%}$ |
| MoFlow | $88.9 \pm 3.2\%$ | $88.0 \pm 2.2\%$ |
| SMILES-LSTM-HC | $33.6 \pm 7.2\%$ | $25.8 \pm 13.5\%$ |
| SELFIES-LSTM-HC | $90.6 \pm 0.6\%$ | $90.5 \pm 1.6\%$ |
| REINVENT | $88.4 \pm 0.1\%$ | $88.3 \pm 0.1\%$ |
| GB-GA | $86.2 \pm 0.4\%$ | $89.2 \pm 0.3\%$ |
| JANUS | $88.2 \pm 0.6\%$ | $88.9 \pm 0.2\%$ |

Table S7: Diversity results for the organic emitters design benchmark objectives. Models are trained on a subset of the GDB-13 dataset. Results are provided as mean and standard deviation of the best objective values from five independent runs in the form mean $\pm$ standard deviation. Metrics: $\Delta E(S_1 - T_1)$: singlet-triplet gap; $f_{12}$: $S_1$ and $S_0$ transition oscillator strength; $\Delta E(S_0\text{-}S_1)$: $S_0$ and $S_1$ vertical excitation energy.

|  | $\Delta E(S_1 - T_1)$ | $f_{12}$ | $+f_{12} - \Delta E(S_1\text{-}T_1) - |\Delta E(S_0\text{-}S_1) - 3.2; eV|$ |
|---|---|---|---|
| SMILES-VAE | $90.1 \pm 1.1\%$ | $78.5 \pm 2.7\%$ | $80.4 \pm 5.3\%$ |
| SELFIES-VAE | $\mathbf{93.0 \pm 0.2\%}$ | $89.5 \pm 0.6\%$ | $92.1 \pm 0.6\%$ |
| MoFlow | $92.2 \pm 1.4\%$ | $89.1 \pm 0.2\%$ | $90.3 \pm 0.6\%$ |
| SMILES-LSTM-HC | $89.90 \pm 0.01\%$ | $89.91 \pm 0.01\%$ | $89.91 \pm 0.01\%$ |
| SELFIES-LSTM-HC | $89.91 \pm 0.01\%$ | $89.91 \pm 0.01\%$ | $89.91 \pm 0.01\%$ |
| REINVENT | $90.8 \pm 0.2\%$ | $90.8 \pm 0.1\%$ | $90.9 \pm 0.1\%$ |
| GB-GA | $92.0 \pm 0.2\%$ | $91.1 \pm 0.6\%$ | $92.1 \pm 0.4\%$ |
| JANUS | $91.7 \pm 0.1\%$ | $\mathbf{92.6 \pm 0.1\%}$ | $\mathbf{92.3 \pm 0.2\%}$ |

Table S8: Diversity metrics for protein-ligand design benchmarks, based on models trained on a subset of the DTP Open Compound Collection. Metrics show mean and standard deviation of optimal target objective values over five independent runs (mean $\pm$ standard deviation). Benchmark metrics: $\Delta E_X$ denotes docking score for protein target $X$.

|  | $\Delta E_{1SYH}$ | $\Delta E_{6Y2F}$ | $\Delta E_{4LDE}$ |
|---|---|---|---|
| SMILES-VAE | $80.2 \pm 0.2\%$ | $91.0 \pm 1.0\%$ | $89.9 \pm 1.4\%$ |
| SELFIES-VAE | $\mathbf{92.8 \pm 0.4\%}$ | $91.9 \pm 0.7\%$ | $90.6 \pm 0.8\%$ |
| MoFlow | $92.5 \pm 1.6\%$ | $\mathbf{92.5 \pm 1.5\%}$ | $\mathbf{92.6 \pm 1.6\%}$ |
| SMILES-LSTM-HC | $91.19 \pm 0.01\%$ | $91.14 \pm 0.04\%$ | $91.15 \pm 0.02\%$ |
| SELFIES-LSTM-HC | $91.223 \pm 0.001\%$ | $91.11 \pm 0.01\%$ | $91.11 \pm 0.01\%$ |
| REINVENT | $92.4 \pm 0.2\%$ | $92.4 \pm 0.3\%$ | $92.4 \pm 0.1\%$ |
| GB-GA | $92.1 \pm 0.3\%$ | $92.2 \pm 0.2\%$ | $92.2 \pm 0.1\%$ |
| JANUS | $91.5 \pm 0.3\%$ | $91.6 \pm 0.7\%$ | $90.9 \pm 1.0\%$ |

Table S9: Diversity results for chemical reaction substrate design benchmarks. Models, trained on a benchmark dataset generated with STONED-SELFIES cycles, yield mean and standard deviation of optimal objective values over five runs (mean $\pm$ standard deviation). Benchmark metrics: $\Delta E^{\ddagger}$: Activation energy of the reaction; $\Delta E_r$: Reaction energy.

| | $\Delta E^{\ddagger}$ | $\Delta E_r$ | $\Delta E^{\ddagger} + \Delta E_r$ | $-\Delta E^{\ddagger} + \Delta E_r$ |
|---|---|---|---|---|
| SMILES-VAE | $70.4 \pm 10.0\%$ | $81.7 \pm 1.6\%$ | $66.5 \pm 7.3\%$ | $64.1 \pm 5.1\%$ |
| SELFIES-VAE | $69.2 \pm 4.4\%$ | $66.9 \pm 5.9\%$ | $64.3 \pm 4.3\%$ | $68.4 \pm 12.6\%$ |
| MOFLOW | $\mathbf{90.6 \pm 1.9\%}$ | $87.1 \pm 3.2\%$ | $\mathbf{84.9 \pm 4.4\%}$ | $\mathbf{85.7 \pm 3.7\%}$ |
| SMILES-LSTM-HC | $34.2 \pm 13.2\%$ | $37.6 \pm 11.2\%$ | $3.8 \pm 1.5\%$ | $18.4 \pm 15.4\%$ |
| SELFIES-LSTM-HC | $59.3 \pm 11.3\%$ | $75.6 \pm 5.4\%$ | $55.6 \pm 19.8\%$ | $72.5 \pm 5.8\%$ |
| REINVENT | $88.0 \pm 0.4\%$ | $\mathbf{88.3 \pm 0.3\%}$ | $83.2 \pm 0.8\%$ | $83.2 \pm 0.8\%$ |
| GB-GA | $81.6 \pm 1.1\%$ | $83.3 \pm 2.2\%$ | $76.3 \pm 1.6\%$ | $80.8 \pm 0.6\%$ |
| JANUS | $77.3 \pm 2.6\%$ | $81.2 \pm 1.4\%$ | $68.9 \pm 4.0\%$ | $71.4 \pm 2.0\%$ |

# S4    Additional Discussion

In this section, we will put our findings into perspective and compare them to the pre-existing knowledge in the field. Specifically, we will address two major talking points. First, we will compare the molecular design benchmarks developed as part of TARTARUS to existing benchmarks that are currently being used. Second, we will investigate the results obtained in the various molecular design objectives in detail and discuss their potential implications for the current status of artificial molecular design and future directions for the field.

## S4.1    Benchmarking with TARTARUS

First, we would like to emphasize that the benchmark results provided as part of this work largely serve as a demonstration of how the objectives can be used to compare the performance of different algorithmic approaches. They should not be viewed as final performance judgement for any of the methods used as model hyperparameters were not optimized comprehensively. Additionally, changing the constraints with respect to computer time and property evaluations will likely lead to different results calling for further efforts towards better benchmark standardization and the introduction of various benchmark scenarios. We believe that explicitly considering the number of evaluations and computation times are still important to make the use of molecular design algorithms more accessible and relevant to the scientific community as long run times are detrimental to widespread application. Nevertheless, we think that useful insights are still to be garnered from our preliminary benchmark results. When looking at all the outcomes combined, one curious observation is that, unlike many previous benchmarking efforts, we do not find one algorithm to perform the best, or at least close to the best, across all the tasks considered. This suggests that the newly introduced design objectives differ significantly in their structure requirements and algorithmic demands. Thus, we believe that different domains of chemistry and material sciences have different structure and design requirements that could potentially motivate the development of algorithms tailored to specific problem domains. At the very least, it suggests that significant further developments are necessary in the field in order to create a true champion algorithm for the design tasks considered.

Next, we will investigate the results of each of the molecular design sets in detail. The benchmark tasks inspired by the design of OPVs prove extremely difficult as most models fail to improve upon the reference structures and, if they do, the improvements are very small. This suggests that the provided dataset already contains almost optimal solutions with respect to the fitness functions chosen. Nevertheless, the models should at least be able to reproduce the dataset results. We suspect that these difficulties of some of the models is likely related to uncertainties in the learned structural space [151]. Notably, the property distributions depicted in Figure S8 show that the multiobjective function is mainly dominated by the PCE values. We believe that increasing the relative importance of the SAscore in the objective function could improve this benchmark task even further. Nevertheless, our results suggest that the design of OPVs based on the Scharber model and an explicit account of synthetic accessibility is delicate and proves challenging for many common molecular design algorithms. Based on the judgement of an expert chemist, the best structures proposed by the molecular design algorithms are likely all stable and synthesizable showing the

importance of accounting for that explicitly in molecular design benchmarks. It is interesting to see that the two best structures found in each of the objectives are quite similar which is likely because they do not stray far from the best compounds already contained in the reference dataset. The best structures for the donor design task seem to prefer having a 2,1,3-thiadiazole moiety while the acceptor design task prefers extended annulated π-systems.

Looking at the objectives based on the design of OLEDs, they are all practically relevant as they are directly based on the design of TADF emitters. The molecular design algorithms showing most consistent performance across the three tasks are JANUS and GB-GA. Interestingly, SELFIES-VAE also shows good performance over all three tasks and it outperforms SMILES-VAE very significantly which could suggest difficulties of the SMILES-VAE to learn the underlying property distribution of the training dataset. Notably, none of the molecular design algorithms used outperform the best molecule in the reference data for the oscillator strength benchmark. Additionally, only three of the models, namely SELFIES-VAE, JANUS, and GB-GA, deliver better properties for the combined objective than the reference structures possess. This demonstrates the difficulty of the corresponding benchmark tasks. Looking at the best performing molecules proposed by the models, the best structures are largely stable and synthesizable. However, there are a few features that are not ideal. In particular, the two molecules with the highest oscillator strength have an oxidized benzene ring that is likely reactive. Additionally, the molecule with the highest combined objective is present in its enol tautomer but it is likely that the corresponding ketone is preferred. This suggests that incorporating the SAscore as a constraint helps increase the stability and synthesizability of the generated molecules but also suggests that additional filters are necessary for higher robustness in that regard. Moreover, some structural features required for favorable properties can be derived. In particular, for high oscillator strength values, extended π-systems are beneficial which is well-known in the field of organic electronic materials [152, 153].

Next, we need to inspect the outcomes of the protein ligand design benchmarks, which also have the lowest number of specific objectives, and is currently the only benchmark set without a multiobjective molecular design task. Most interestingly, while the general setup of all the three objectives is identical, we find the performance of the molecular design algorithms to be varying significantly depending on the specific target. This suggests that designing ligands based on molecular docking for the three proteins chosen differs significantly in terms of structural requirements. Most models succeed at proposing ligands with better docking scores than the best reference compounds for all three target proteins. In terms of the constraints, surprisingly, the VAEs show the poorest performance which hints towards difficulties during training. Otherwise, JANUS also generally shows somewhat lower success rates which is likely caused by its limited features to bias structure generation by reference compounds and its tendency to explore more diverse structural regimes to potentially find new candidate solutions. This is relevant as the success rates are not only measured on the best-performing molecules but over all the structures proposed. In contrast, the LSTM-HC models, REINVENT and GB-GA consistently reach higher success rates demonstrating that they succeed at learning to propose desirable structures based on the reference dataset. Additionally, it likely also implies that exploration far outside of the distribution of the reference dataset is avoided. Thus, overall, REINVENT and GB-GA provide the most promising results in these protein ligand design benchmarks. When looking at the best-performing structures, it is notable that molecules resembling steroids seem to be good binders for at least two of the protein targets selected. Notably, there is one important aspect to be discussed that relates specifically to the dataset of reference compounds and the simulation workflow. While the DTP Open Compound Collection is particularly attractive to be used as training dataset because all these structures have been synthesized and tested experimentally for potential use as drug candidates, it suffers a major drawback in terms of data quality. Unfortunately, the majority of structures lack proper definition of stereochemistry despite often having multiple stereogenic centers. While our computational workflow is deterministic with respect to the particular stereoisomer produced when provided with SMILES without defined stereochemistry, it is still very likely that stereoisomers were produced and simulated that do not correspond to the ones of the actual molecules in the compound collection. Thus, when SMILES without well-defined stereochemistry are submitted to the property prediction workflow, the 3-dimensional structure generated in the process needs to be inspected to derive the stereoisomer computed and guide subsequent experimental validation. For even more realistic drug design benchmarks tasks in the future, a reference dataset of structures that have both been experimentally synthesized before and

have well-defined stereochemistry that reflects the experimental samples needs to be used.

When investigating the results of the chemical reaction substrate design benchmark tasks, we find that only the two GAs employed consistently outperform the best reference structures. This is particularly notable as both JANUS and GB-GA achieve large improvements over the best properties in the training set for all but the fourth benchmark objective showing that there is, in principle, significant room for further improvement. Additionally, in many cases, the proposed molecules perform significantly worse than the best-performing reference compounds suggesting that many of the algorithms used seem to have difficulties to learn from the provided dataset. This could potentially point towards peculiarities of the provided training set as it was the only one that was created in a random way from the necessary core structure. It might be too small in size paired with a high diversity of structural moieties making it challenging to extract structure-property relationships. Notably, as expected because it goes against the Bell–Evans–Polanyi principle, the fourth benchmark objective proves to be extremely difficult with the best results providing only comparably small improvements over the reference set. The structures of the best proposed compounds for each of the design objectives provide some preliminary insights into the structural features effecting favorable properties (cf. Figure S4). The sites donating hydrogen atoms seem to prefer being more substituted for both lowering the barriers and lowering the reaction energies. This effect can be understood by a reduction of steric repulsion between these substituents and the bridge substituents when going from tetrahedral to trigonal geometry around the respective carbon atom. Particularly effective for lowering reaction energies seem moieties suitable for conjugation with the newly generated double bond in the product, owing to the stabilizing effect of extended conjugation. To break the correlation between reaction energies and activation energies, the introduction of two mesomerically electron-donating groups appear most suitable, potentially due to the opposing factors of sterics, lowering the reaction energy as outlined above, and electronics. Notably, deciphering substituent effects for these type of dyotropic reactions is known to be difficult [154], and would require more extensive and systematic analysis and corroboration which is outside the scope of this work.

The final benchmark of TARTARUS to be discussed is the model timing comparison. This is important as it can provide information about whether improved optimization performances originate from the utilization of more computational resources and longer training times. It should be noted that the absolute computation times reported are not useful as they strongly depend on the computer hardware available. Hence, we recommend them to be used as a relative rather than an absolute measure. Thus, when new models are being tested for timing, some of the molecular design algorithms provided here as a reference should be run again with the same hardware as internal standards permitting a relative timing comparison to all the other algorithms. Looking at the timing results, apart from the VAEs, we consider training to be still affordable for the deep generative models tested, even without the availability of GPUs. However, without GPUs, training the VAEs is computationally too expensive and is detrimental to its widespread application, and even for the LSTM-HC models it is comparably cumbersome. Additionally, while molecule sampling time, taking roughly on the order of a few minutes in most cases, is generally reasonable, we believe that there is considerable room for further improvement. This is particularly evident from the extremely good timing performance of REINVENT taking less than one minute for that step. Nevertheless, as it is more important to propose several meaningful structures than to generate a large number of structures fast, we believe that the sampling times of the all the algorithms tested are sufficient for practical purposes.

## S5 Computational Details

### S5.1 Molecular Design Algorithms

#### S5.1.1 VAE

We implemented a variational autoencoder (VAE) architecture from the literature [31]. In particular, the encoder used three 1D convolutional layers with filter sizes of 9, 9, and 10 convolution kernels, followed by one fully connected layer that encodes a latent space of size 292. The decoder fed into three layers of gated recurrent unit (GRU) networks [107] with a hidden dimension of 501 each. Based on the respective training dataset to be used and the encoding used for representing

the molecules (i.e., SMILES or SELFIES), we modified the alphabet size for converting the list of molecules into one-hot encodings. Additionally, the string length of the largest molecule from the dataset is used to pad sequences to the same length. Based on the dataset, these two parameters lead to minor modifications in the sizes of the first and last layers of the encoder and decoder, respectively. For each dataset, the models are trained using 24 CPUs and a single GPU, with training times ranging between 2 and 10 hours. Generally, 80% of the reference molecules are used for training, while the remaining ones are used for testing. We manually optimize the learning rate, learning rate decay rate, number of epochs for training, and the batch size based on performance on the test set. After obtaining a fully trained model, we perform structure optimization. The best known molecule for a given task, i.e., the one with the most favorable score, is converted into a one-hot encoding and passed through the encoder to the latent space. Subsequently, Gaussian noise is added to the latent space vector to produce a population of vectors. These vectors are passed through the decoder to produce molecules that are evaluated via the respective objective function. The best molecule resulting from this population is then used to repeat this procedure.

### S5.1.2   JANUS

Throughout the benchmarks, we used the default implementation of JANUS (`https://github. com/aspuru-guzik-group/JANUS`). We used version 1.0.3 of SELFIES, and the default valence constraints implemented in that version of SELFIES were employed. Additionally, no structure filters were employed in the genetic operators. For initiation, the file `sample_start_smiles.txt` was populated with the 1,000 best molecules from the corresponding benchmark reference dataset. The `generation_size` parameters and the other parameters that influence the generations are modified based on the respective benchmark task. An artificial neural network (ANN) classifier is used for additional selection pressure for all the tasks (cf. JANUS+C in the original publication [7]).

### S5.1.3   GB-GA

We employed of the implementation provided by Jensen [34] in the following repository: `https: //github.com/jensengroup/GB_GA/`. For initiation, the file `ZINC_first_1000.smi` was populated with the 1,000 best molecules from the corresponding benchmark reference dataset for pre-conditioning to derive both mutation and crossover probabilities. The file `scoring_functions.py` was modified each time to incorporate the corresponding fitness functions from different benchmark tasks.

### S5.1.4   LSTM-HC

The long short-term memory hill climbing (LSTM-HC) approach uses a pre-trained long short-term memory (LSTM) [105] recurrent neural network (RNN) to generate sequences of molecular strings [33], and can be implemented with either SELFIES or SMILES. For each molecule, the string representation characters are transformed into one-hot encoded vectors, which are padded up to the length of the longest string in the respective reference dataset for each benchmark task. Similar to how it was implemented in the literature [33], the model consisted of 3 stacked LSTMs, each with 1024 hidden dimensions and a dropout ratio of 0.2, followed by a linear layer that outputs the logits of the next predicted character in the string sequence. That way, the output returns the probability for observing a certain subsequent character, given the preceding characters in the sequence. The model was pre-trained with a batch size of 128, using the ADAM optimizer [155] with a learning rate of 0.001. For optimization in each of the benchmark tasks, the top 2 known molecules with the best fitness values are used to create seeds for the LSTM structure generation. For SMILES, the strings are truncated randomly at 25% to 75% of the size of the original sequence. For SELFIES, the strings are first randomized 5 times using reordered SMILES that represent the same molecule before they are being truncated. Subsequently, the truncated strings are passed into the model and the next characters are generated by sampling from the probability distribution that is generated by the LSTM output. Each iteration, the model generates 500 new molecules for evaluation, and the top 2 molecules are selected for the subsequent iteration. Additionally, the model is re-trained after each iteration on the set of new molecules for 10 epochs at a lower learning rate of 0.0001.

### S5.1.5 MoFlow

We employed the MoFlow framework developed by Zang et al. [32] for our molecular structure generation tasks. The implementation we used is publicly accessible at `https://github.com/calvin-zcx/moflow`. We trained the model on 80% of the respective dataset, using a batch size of 256 for a maximum of 200 epochs. All model parameters were selected based on the example provided in the GitHub repository. Specifically, the B-Block of the model utilized 10 flows and had 512 hidden channels in both layers. In the A-Block, the model was configured with 38 flows and had 256 hidden graph neural network channels, along with hidden linear layers of sizes 512 and 64. Masking parameters were set with both a row size list and a row stride list of 1. A noise scale of 0.6 was employed to introduce a stochastic element into the behavior of the model. To optimize properties, we implemented a hill-climbing-based algorithm. In this approach, the model was randomly sampled, and the top 250 molecules were selected from a total of 500 oracle calls for subsequent training. This cycle was repeated for up to 10 iterations, depending on the benchmarking task.

### S5.1.6 REINVENT

In REINVENT [30], similar to LSTM-HC, an LSTM that is pretrained on a reference dataset is used as a prior for generating new SMILES sequences. Reinforcement learning (RL) is used to optimize for a particular fitness value. The LSTM model acts as an agent, where the addition of a character is an action, and the output probability distribution is the action probability, or the policy. The action probability is augmented by a scoring function $S(m) \in [-1, 1]$, with larger scores corresponding to more desirable fitness values. SMILES that do not correspond to valid molecules are assigned a score of 0. The algorithm updates the policy to increase the expected fitness while still keeping the learned conditional sequence probabilities to produce valid SMILES that resemble the structural distribution of the reference dataset. The scoring function is a modified sigmoid function parameterized by the known fitness values from previous iterations. Let the set of known fitness values be $F$. The scoring function for a molecule $m$ is defined as

$$S(m) = \frac{2}{1 + e^{-b(f-a)}} - 1, \tag{2}$$

where $a$ is the average of $F$, and $b$ is the slope of the sigmoid. This sigmoid is defined as

$$b = -\frac{1}{max(F) - a} \ln \left( \frac{2}{c+1} - 1 \right), \tag{3}$$

where $c$ is a threshold value set to 0.8. This means for the current fitness values in $F$ that the best known fitness will map to 0.8 in the scoring function. The smaller the threshold $c$, the stronger the reward for fitness values larger than the best one contained in $F$. Further details about the approach and model parameters are provided in the original paper [30].

## S5.2 Benchmarks

### S5.2.1 Design of Organic Photovoltaics

**Workflow**    To set up the property prediction workflow, we first implemented the Scharber model [41]. To do that, we constructed a simplified model to predict the short-circuit current density ($J_{SC}$). The canonical way to do that is via definite integration of the spectrum of the light source over the energy range of the absorbed light of the simulated device [42]. To bypass having to perform the integration in the property simulation workflow and provide the Reference Air Mass 1.5 Spectra (AM1.5G) as part of TARTARUS, we created a regression model for the short-circuit current density as a function of the band gap energy $E_G$. We used the AM1.5G spectrum that accounts for both direct and circumsolar irradiation and performed the integration via the trapezoidal rule as implemented in the `numpy` package. We found that a simple Gaussian function of the form

$$J_{SC} = A \cdot e^{-\frac{E_G{}^2}{B}}, \tag{4}$$

where $A$ and $B$ are fitting parameters, provides an excellent fit to the actual integration as demonstrated in the comparison between the original relationship derived from integration of the AM1.5G reference data (denoted "Original") and the fitted function (denoted "Fit") in Figure S6. Importantly, the regression model assumes an external quantum efficiency of 0.65, which is typically used in the Scharber model [41]. Typically, the band gap energy is approximated by the gap between the HOMO of the donor and the LUMO of the acceptor, which then, in conjunction with the regression model just described, allows to estimate the short-circuit current density that is needed to compute the power conversion efficiency (PCE) under the assumptions of the Scharber model [41].

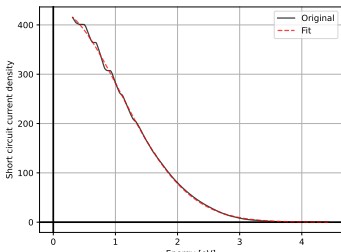

Figure S6: Short circuit current density as a function of the band gap energy of the light-absorbing material in an organic solar cell. The original dependency results from the integration of the Reference Air Mass 1.5 Spectra (AM1.5G). The fit depicts the result of regression with a Gaussian function with two fitting parameters.

Besides the computation of the short-circuit current density, the procedure described in the original paper was followed [41]. The open circuit voltage ($V_{OC}$) is determined as energy difference between HOMO of the donor and the LUMO of the acceptor minus 0.3 eV of overpotential that is typically assumed to be required for any charge separation to take place [42]. Should the open circuit voltage be formally estimated to be negative then it is assumed to be 0. In case the LUMO of the donor is less than 0.3 eV higher in energy than the LUMO of the acceptor then the open circuit voltage is also assumed to be 0. The power conversion efficiency is estimated based on the equation

$$PCE = 100\% \cdot V_{OC} \cdot FF \cdot \frac{J_{SC}}{P_{in}}, \tag{5}$$

where $V_{OC}$ is the open circuit voltage , $FF$ is the fill factor, $J_{SC}$ is the short-circuit current density, and $P_{in}$ is the incident light intensity obtained by integration over the entire AM1.5G reference spectrum and therefore is constant. In the Scharber model, the fill factor is typically assumed to be optimizable to reach a value of 0.65 [156], which is thus inserted into this equation. For the first task of the organic photovoltaic design benchmarks, the goal is to design a small organic donor molecule used with [6,6]-phenyl-C61-butyric acid methyl ester (PCBM) as acceptor. The LUMO energy level of PCBM is -4.3 eV and all light is assumed to be absorbed by the donor molecule. Notably, this is a common design objective that has been pursued in the OPV literature [41, 42] and was also implemented in the Harvard Clean Energy Project (CEP) [37, 40]. For the second task, a small organic acceptor molecule is to be designed that is used with poly[N-90-heptadecanyl-2,7-carbazole-alt-5,5-(40,70-di-2-thienyl-20,10,30-benzothiadiazole)] (PCDTBT) as a donor. The HOMO energy level of PCDTBT is -5.5 eV and all light is assumed to be absorbed by the acceptor molecule. This benchmark task is based on a high-throughput virtual screening that was conducted in the literature to find new non-fullerene acceptors [43].

The computational property prediction workflow accepts a SMILES string as input and uses `Open Babel` [157] to generate initial guess Cartesian coordinates. Next, `crest` [158] is used for conformer search of the molecule via the iMTD-GC workflow at the GFN-FF level of theory [159] using the additional keywords `-mquick`, and `-noreftopo`. Subsequently, the lowest energy conformer resulting from this search is used for a geometry optimization at the GFN2-xTB level of theory [54]. The properties of interest, i.e. HOMO energy, LUMO energy, HOMO-LUMO gap and molecular dipole moment, are obtained from the minimized structure at the same level of theory.

Importantly, to obtain better PCE estimates, we calibrated the HOMO and LUMO energies obtained from GFN2-xTB simulations against the corresponding density functional approximation (DFA) results taken from the Harvard Clean Energy Project Database (CEPDB) for the reference molecules of the CEPDB subset. In particular, the median HOMO and LUMO energies of all the DFA results contained in the CEPDB were used. Both the HOMO and the LUMO energies were calibrated against the respective medians of the DFA HOMO and LUMO energies using a Theil-Sen estimator [160, 161], as implemented in the python package `scikit-learn` [162], due to its robustness with respect to outliers. The maximum number of subpopulations set to 250,000. The linear regression functions derived are illustrated in Figure S7 and are provided in the following equations:

$$E_{HOMO,calibrated} = E_{HOMO,GFN2-xTB} \cdot 0.8051 + 2.5377 \, eV \tag{6}$$

$$E_{LUMO,calibrated} = E_{LUMO,GFN2-xTB} \cdot 0.8788 + 3.7913 \, eV \tag{7}$$

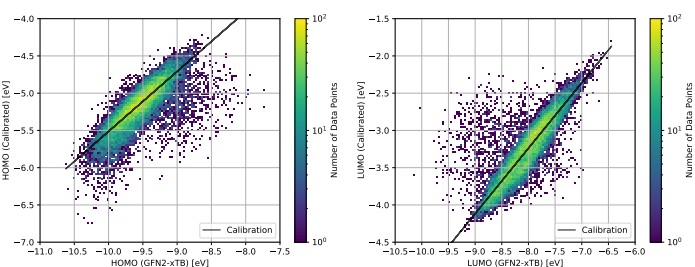

Figure S7: Two-dimensional histograms depicting the comparison of HOMO (left) and LUMO (right) energies obtained using DFA computations taken from the CEPDB against the corresponding energies at the GFN2-xTB level of theory obtained using the property simulation workflow developed for the design of organic photovoltaics benchmarks. The black lines were obtained via Theil-Sen estimators for linear regression.

As can be seen in Figure S7, the assumption of a linear relationship between the energies at the GFN2-xTB level of theory and at the DFA level of theory is reasonable.

**Dataset** The CEPDB subset used for training all the generative models was obtained from the following repository: `http://github.com/HIPS/neural-fingerprint`. For all molecules in this dataset, we simulated the corresponding properties of interest with the developed property simulation workflow (cf. Figure S8).

### S5.2.2 Design of Organic Emitters

**Workflow** The computational property prediction workflow accepts a SMILES string as input and uses both `Open Babel` [157] and `RDKit` [163] to generate initial guess Cartesian coordinates. Whichever guess coordinates have a lower in energy at the GFN-FF level of theory are used as starting point for the subsequent conformer search of the molecule via the iMTD-GC workflow at the GFN-FF level of theory [159] using the additional keywords `-mquick`, and `-noreftopo`. Subsequently, the lowest energy conformer resulting from this search is used for a geometry optimization at the GFN0-xTB level of theory [53, 63, 64]. Afterwards, the optimized geometry is used to perform a TD-DFT single point calculation at the B3LYP/6-31G* level of theory [65–70] using the PySCF python package (version 1.7.6) [164–166]. Density fitting is utilized via the `def2-universal-jkfit` auxiliary basis set. For the excited state simulation, two excited states are generated for both the singlet and the triplet manifold. Notably, in case the molecule contains iodine, the LANL2DZ basis set [167] is used for that atom. From the corresponding results, the energies of the lowest excited singlet and triplet states, and the oscillator strength for the transition between the ground state and the lowest excited singlet state are obtained. The singlet-triplet gap is derived as energy difference between the first

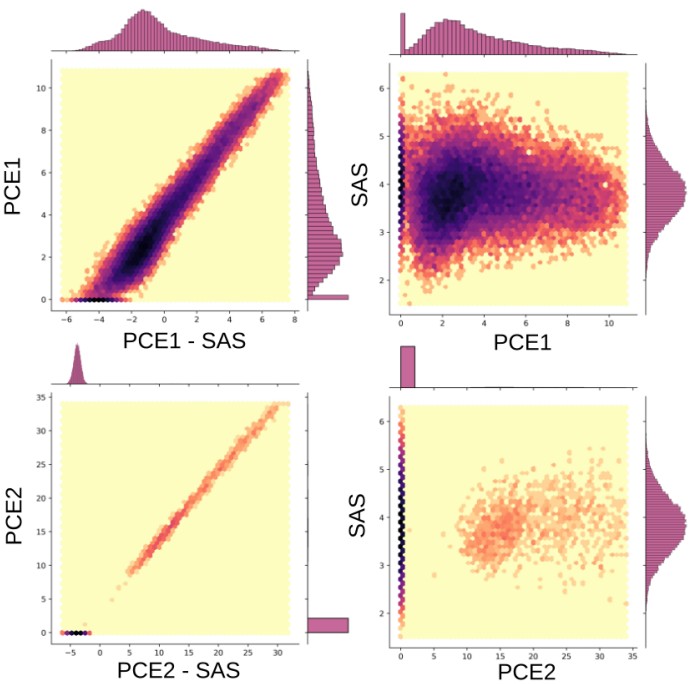

Figure S8: Two-dimensional histograms depicting the property distributions of the provided reference dataset as obtained from the developed simulation workflow for the design of organic photovoltaics benchmark set. Top left: multiobjective target function $PCE_{PCBM} - SAscore$ against power conversion efficiency with PCBM as acceptor $PCE_{PCBM}$; top right: synthetic accessibility metric $SAscore$ against power conversion efficiency with PCBM as acceptor $PCE_{PCBM}$; bottom left: multiobjective target function $PCE_{PCDTBT} - SAscore$ against power conversion efficiency with PCDTBT as donor $PCE_{PCDTBT}$; bottom right: synthetic accessibility metric $SAscore$ against power conversion efficiency with PCDTBT as donor $PCE_{PCDTBT}$.

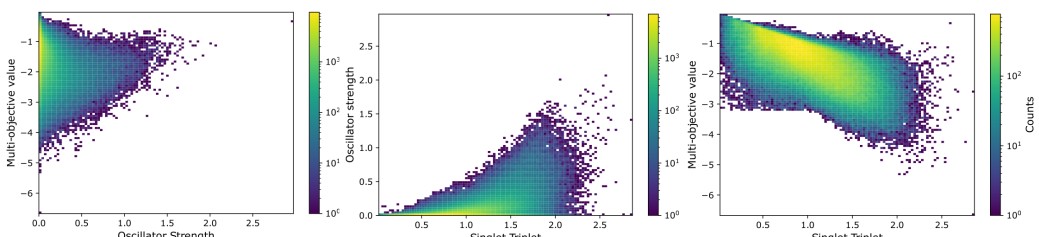

Figure S9: Two-dimensional histograms depicting the property distributions of the provided reference dataset as obtained from the developed simulation workflow for the design of organic emitters benchmark set. Left: multiobjective target function $+f_{12} - \Delta E(S_1\text{-}T_1) - |\Delta E(S_0\text{-}S_1) - 3.2\ eV|$ against oscillator strength for the transition between $S_1$ and $S_0$, $f_{12}$; middle: multiobjective target function $+f_{12} - \Delta E(S_1\text{-}T_1) - |\Delta E(S_0\text{-}S_1) - 3.2\ eV|$ against singlet-triplet gap, $\Delta E(S_1 - T_1)$; right: oscillator strength for the transition between $S_1$ and $S_0$, $f_{12}$ against singlet-triplet gap, $\Delta E(S_1 - T_1)$.

excited singlet and the first excited triplet state. Notably, only molecules with an SAscore below or equal to 4.5 are subjected to the property simulation workflow. All molecules above are assigned a very unfavorable fitness.

**Dataset**  We developed a comprehensive set of filters to define a subset of GDB-13 [62], originally comprising more than 970 million organic molecules, with over 400,000 structures possessing cycles and a high degree of conjugation, that we herein refer to as GDB-13$_{SUB}$. These filters ensure that the molecules encompass cyclic $\pi$-systems to increase representation of the relevant structural space for potential organic emitters. They were implemented via RDKit [163] and are summarized in Table S10. For the forbidden substructures, we utilized the following SMARTS patterns:

```
[Cl,Br,I], *=*=*, *#*, [O,o,S,s]~[O,o,S,s],
[N,n,O,o,S,s]~[N,n,O,o,S,s]~[N,n,O,o,S,s],
[C,c]~N=,:[O,o,S,s;!R],
[N,n,O,o,S,s]~[N,n,O,o,S,s]~[C,c]=,:[O,o,S,s,N,n;!R],
*=[NH], *=N-[*;!R], *~[N,n,O,o,S,s]-[N,n,O,o,S,s;!R],
*-[CH1]-*, *-[CH2]-*, *-[CH3]
```

Table S10: List of filters employed to create the $\pi$-systems subset of GDB-13, GDB-13$_{SUB}$. In each line, $x$ denotes the value of the corresponding feature.

| Number | Feature | Definition | Value |
|---|---|---|---|
| 1 | Charge | Charge of the molecule. | $x = 0$ |
| 2 | Radicals | Number of radical electrons. | $x = 0$ |
| 3 | Bridgehead Atoms | Number of bridgehead atoms. | $x = 0$ |
| 4 | Spiro Atoms | Number of spiro atoms. | $x = 0$ |
| 5 | Aromaticity Degree | Percentage of aromatic non-hydrogen atoms. | $x \geq 0.5$ |
| 6 | Conjugation Degree | Percentage of conjugated bonds between non-hydrogen atoms. | $x \geq 0.7$ |
| 7 | Maximum Ring Size | Size of the largest ring. | $4 \leq x \leq 8$ |
| 8 | Minimum Ring Size | Size of the smallest ring. | $4 \leq x \leq 8$ |
| 9 | Substructures | Presence of any forbidden substructures (see text). | FALSE |

After filtering the GDB-13 dataset with these filters, and removing all molecules with an SAscore larger than 4.5, we subjected the resulting molecules to the property simulation workflow described above to obtain the reference dataset for the design of organic emitters benchmarks. The corresponding property distributions are visualized in Figure S9.

### S5.2.3  Design of Protein Ligands

**Workflow**  To set up the computational property prediction workflow with docking calculations, we downloaded crystal structures for the three proteins of interest from the Protein Data Bank (PDB). The corresponding PDB codes are 1SYH, 4LDE, and 6Y2F. The corresponding structures were all co-crystallized with a ligand bound to the protein. All these structures were processed using the protein preparation wizard [168] as implemented in the computer program Schrödinger (version 12.9). We created a box around the bound ligand for each of the protein structures based on re-docking accuracy. Subsequently, the native ligand was removed from the structure providing the starting setup for new ligands to be bound in the property simulation workflow. This workflow accepts a SMILES string as input, converts it into an initial 3D guess geometry using `Open Babel` [169] and exports it as a `PDBQT` file. Subsequently, the quality of the molecules is checked by running `obenergy`. During reading of the SMILES, the corresponding structure is inspected for the presence of the following SMARTS patterns:

```
*1=**=*1, *1*=*=*1, *1~*=*1, [F,Cl,Br]C=[O,S,N], [Br]-C-C=[O,S,N],
[N,n,S,s,O,o]C[F,Cl,Br], [I], [S&X3], [S&X5], [S&X6],
[B,N,n,O,S]~[F,Cl,Br,I], *=*=*=*, *=[NH], [P,p]~[F,Cl,Br], SS, C#C, C=C=C,
*~[P,p](=O)~*, C=C=N, NNN, "[*;R1]1~[*]~[*]~[*]1", OOO,
```

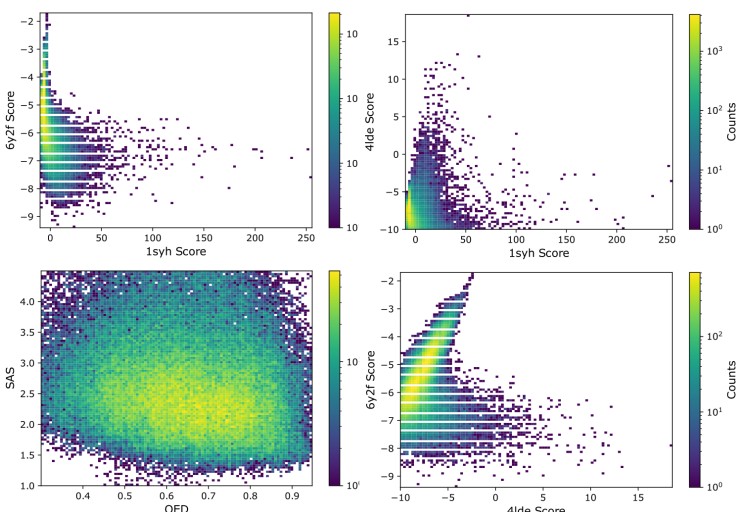

Figure S10: Two-dimensional histograms depicting the property distributions of the provided reference dataset as obtained from the developed simulation workflow for the design of protein ligands benchmark set. Top left: docking score to 4LDE $\Delta E_{4LDE}$ against docking score to 1SYH $\Delta E_{1SYH}$; top right: docking score to 6Y2F $\Delta E_{6Y2F}$ against docking score to 1SYH $\Delta E_{1SYH}$; bottom left: docking score to 6Y2F $\Delta E_{6Y2F}$ against docking score to 4LDE $\Delta E_{4LDE}$; bottom right: synthetic accessibility metric $SAscore$ against quantitative estimate of drug-likeness $QED$.

```
[#8]1-[#6]2[#8][#6][#6][#6]12, N=C=O, C1CN1, [#6](=#8])[F,Cl,Br,I],
[#6](=[#8])=[#6](-[#8])-[#6](=[#8])~[#8], N(-[#6])=[#7]-[#8].
```

These SMARTS patterns were developed from our experience with the use of docking in conjunction with inverse molecular design algorithms. They correspond to reactive structural moieties that can lead to very favorable docking scores and should therefore be explicitly avoided [7]. Additionally, it is also tested whether the structure is charged, possesses radical electrons, contains more than two bridgehead atoms, has at least one ring with more than 8 members and violates Lipinski's Rule of Five [128]. If at least one of these initial checks is true, the property simulation workflow is aborted and a very bad fitness of 10,000 is returned. If all of these checks are false, the workflow is continued. Subsequently, `smina` is used to add the ligand to the prepared protein structure of interest and perform a docking simulation with an exhaustiveness setting of 10. The docked ligand structure with the lowest docking score is selected and the corresponding docking score returned by the simulation workflow. Following recent literature,[84, 85] the quality of the generated docked structure was checked using `obenergy` and only values below 10,000 were accepted.

**Dataset**   We obtained a list of approximately 250,000 canonical SMILES from PubChem [170] that are part of the DTP Open Compound Collection [79, 80]. Next, all the structures that do not pass any of the structural filters described in the previous section are removed. Thus, we obtained a set of 152,296 molecules that are provided as reference dataset for the protein ligand design benchmark set and used it for training the generative models. For all molecules in this dataset, we simulated the corresponding properties of interest with the developed property simulation workflow (cf. Figure S10).

### S5.2.4  Design of Chemical Reaction Substrates

**Workflow**   The double hydrogen transfer in *syn*-sesquinorbornenes (cf. Figure S11A) was chosen as a suitable benchmark reaction due to its robust behavior in simulations. Additionally, it is well-described by the SEAM method as shown previously by Jensen [89]. For the benchmark, we allow

substitutions at the positions indicated by **X** or **R** in Figure S11B.

Figure S11: Intramolecular concerted double hydrogen transfer reaction of *syn*-sesquinorbornenes (A) and positions in the *syn*-sesquinorbornene core structure allowed to be substituted (B).

The general procedure of the reactivity workflow we developed is outlined in the following steps:

1. Generation of initial conformation of the reactant.
   (a) Try to generate coordinates with `Open Babel` [157].
   (b) In case of failure, generate coordinates with `RDKit` [163].
2. MMFF [171] optimization of the reactant.
3. `crest` [158] conformer search for the reactant via the iMTD-GC workflow.
4. Constrained MMFF optimization going from the reactant to the product in conformation consistent with reactant.
5. Full MMFF optimization of the product.
6. `crest` conformer search for the product lowest conformation via the iMTD-GC workflow.
7. Crude reaction path interpolation with the geodesic interpolation method [172].
8. SEAM [89] optimization of the TS.
9. Constrained `crest` conformer search of the TS via the iMTD-GC workflow.
10. SEAM re-optimization of the lowest energy TS conformer.
11. Calculation of reaction and activation energy from the SEAM model.

Prior to performing any property simulations, the molecule is checked for the presence of the *syn*-sesquinorbornene core motif which is strictly required for the reaction of interest to occur:

```
[H][C@@]1(*)[C@;R2](*)2[C@@]34[C@;R2]5(*)[C;R1](*)=[C;R1](*)[C@;R2](*)
([*;R2]5)[C@@]3([*;R1]4)[C@](*)([*;R2]2)[C@@;R1]1([*])[H]
```

Additionally, the input structure is inspected to ensure the absence of several reactive structural moieties as described by the following SMARTS patterns:

```
[C-],  [S-],  [O-],  [N-],  '[*+],  [*-]   [PH],  [pH],  [N&X5],
*=[S,s;!R],  [S&X3],  [S&X4],  [S&X5],  [S&X6],  [P,p],
[B,b,N,n,O,o,S,s]~[F,Cl,Br,I],  *=*=*,  *#*,  [O,o,S,s]~[O,o,S,s],
[N,n,O,o,S,s]~[N,n,O,o,S,s]~[N,n,O,o,S,s],
[N,n,O,o,S,s]~[N,n,O,o,S,s]~[C,c]=,
:[O,o,S,s,N,n;!R],  *=N-[*;!R],  *~[N,n,O,o,S,s]-[N,n,O,o,S,s;!R]
```

A detailed description of some of the steps in the property simulation workflow for the design of chemical reaction substrates is provided in the following bullet points:

1. SEAM optimization of the TS:
   The SEAM model optimizes the transition state as the crossing point between reactant and product force fields. In this work, we used the GFN-FF force field [159], and shifted the product force field to reproduce the energy difference between the reactants and products at the GFN2-xTB level of theory [54]. For the product force field shift, we used the lowest

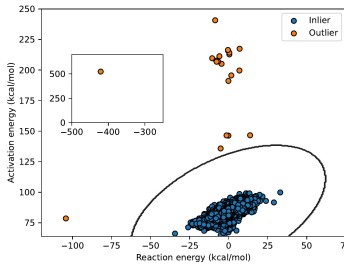

Figure S12: Illustration of the outlier detection cutoff for the elliptic envelope used for the property simulation workflow of the design of chemical reaction substrates benchmark set. Data points appearing outside the black line are regarded as outliers and assigned a very low fitness.

energy conformers of the reactant and product, respectively. The optimization was carried out on the upper SEAM surface by introducing a small coupling of 0.001 eV between the states. The guess structure for the TS was taken from a geodesic interpolation between reactant and product geometries [172]. The TS itself was optimized with the geomeTRIC package [173] via an interface to `pyscf` [164–166]. The subsequent `crest` transition state conformer search used constrained C—H—C bond lengths with a force constant of 1.0 a.u. in strength. All calculations were carried out using the POLANYI package, which will be reported in detail elsewhere. The activation energy was taken as the difference of the GFN-FF energy of the optimized TS structure and the energy of the lowest energy conformer of the reactant. The reaction energy was taken as the energy difference between the lowest energy conformers of the product and reactant, including the energy shift of the product force field. These energies are electronic energies from the force field. While free energies would be preferable, it would increase the complexity of the workflow, and significant error cancellation is expected for this intramolecular reaction.

2. `crest` conformer searches:
   We opted to perform the `crest` conformer searches with the keywords `-mquick`, `-gfn2//gfnff`, and `-noreftopo`. Although it would be more consistent to use `-gfnff`, we found that it produced a larger number of outliers in preliminary test calculations.

3. Outlier detection:
   Although the workflow is very robust, it does produce some outliers due to failures in the conformational search or the TS optimization. In the worst cases, these outliers could display extreme properties such as artificially low or high reaction or activation energies, and, hence, sometimes emerge as the best performering structures. One solution is increasing the exhaustiveness of the conformational sampling, but this also leads to significantly longer computational run-times. Instead, we opted for an outlier detection model based on the computed reaction and activation energies. According to the Bell-Evans-Polanyi relationship [148], the reaction energy is expected to correlate with the activation energy. Therefore, an outlier model based on the 2D scatter plot of these two properties can be used to identify potential outliers. We used the `EllipticEnvelope` from scikit-learn [174]. The `contamination` hyperparameter was set to 0.00035 by visual inspection to allow for a considerable deviation from the correlation observed in the reference dataset, while simultaneously capturing clear outliers. This outlier detection scheme is illustrated in Figure S12. Accordingly, we incorporated this outlier detection into the property simulation workflow and assigned an extremely low fitness of $-10^4$ to outlier molecules.

**Dataset** We constructed a dataset of 60,850 molecules by performing multiple mutations of various SELFIES representations of the parent *syn*-sesquinorbornene. Specifically, the starting structure was used to produce 20 randomly reordered SMILES strings. The resulting SMILES were converted to SELFIES and 20 random mutations were conducted via STONED-SELFIES [112]. All mutants were subsequently converted to SMILES and checked for the presence of the core motif. Mutations that lead to structures without the core motif were discarded. Additionally, molecules that did not pass the filters were also removed. This procedure was repeated on all unique and valid mutated structures until the final number of molecules was reached. Notably, we used the default mutation settings of STONED-SELFIES [112]. When running the dataset through the property simulation workflow, only

35 molecules either did not run through the property simulation workflow properly or were outliers, corresponding to a success rate of 99.94%. The results of running the reference structures through the property simulation workflow with respect to both property distributions and computational run time are illustrated in Figure S13.

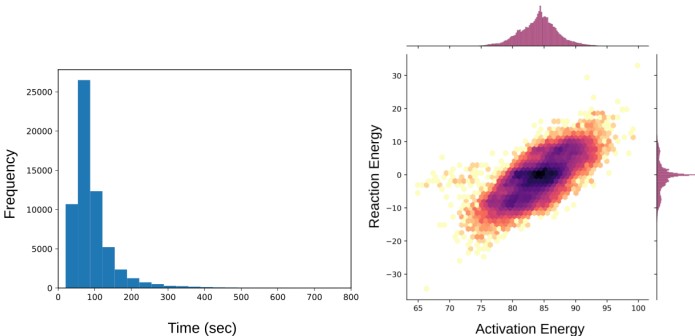

Figure S13: Simulation results for the reference structures in the developed simulation workflow for the design of chemical reaction substrates benchmark set. Left: Histogram of the simulation workflow run times; right: two-dimensional histograms depicting the property distributions of the provided reference dataset: reaction energy $\Delta E_r$ against activation energy $\Delta E^{\ddagger}$.

## S6 Model Timing Comparison

All deep generative models (SMILES-VAE, SELFIES-VAE, MoFlow, REINVENT, SMILES-LSTM-HC and SELFIES-LSTM-HC) were trained using the first 80% of the molecules from the protein ligand design tasks. The remaining 20% of the dataset was used to assess convergence and perform manual hyperparameter optimization. The corresponding figure in the main text provides four timing benchmark metrics: (1) Training Time with GPU, (2) Sample time for 10,000 molecules, (3) Epoch time without GPU and (4) Epoch time with GPU. For each metric, 5 independent experiments were conducted to obtain both average and standard deviations of the respective values. For the first metric, all the generative models were trained using 24 CPUs (AMD Rome 7532  2.40 GHz 256M cache L3) and a single GPU (Tesla A100). For REINVENT, MoFlow, and the LSTM-HC models, a batch size of 512 was used for training. In contrast, the VAE models performed better in terms of validity and reconstruction with a smaller batch size of 8. For sampling times of 10,000 molecules, we used fully trained deep learning models as obtained after the training time measurements and determined the time required for generating 10,000 molecules in the presence of 24 CPUs (AMD Rome 7532 @ 2.40 GHz 256M cache L3) and a single GPU (Tesla A100). The batch size was 512 during sampling for all deep generative models. For GAs, the molecule sampling time was measured only in the presence of 24 CPUs as both JANUS and GB-GA lack GPU support. Notably, in the models that use SELFIES as a molecular representation, the time needed for translating SELFIES to SMILES was included in the final reported time. For JANUS and GB-GA, this metric was obtained by increasing the population size to 10,000 and by using single generation runs and a dummy fitness function that returns a value of 1 for any molecule. For the single epoch times, we report the time required for training models in the presence of either 24 CPUs (AMD Rome 7532 @ 2.40 GHz 256M cache L3) or 24 CPUs (AMD Rome 7532 @ 2.40 GHz 256M cache L3) with a GPU (Tesla A100). For the deep generative models, a batch size of 512 was used. Notably, for both GB-GA and JANUS, the CPU epoch time corresponds to the total time the models need to precondition the genetic operators based on a dataset of reference structures. Additionally, for both these models, preconditioning was performed by providing 1,000 random SMILES from the reference dataset of the protein ligand design task to the molecular design algorithms. Finally, using the CPU epoch times and the number of epochs needed for training when also using a GPU, we estimated the total CPU training times for all the deep generative models by multiplying these two numbers. Notably, we decided not to estimate the corresponding estimated errors via error propagation as these values are merely rough estimates.

