# OpenReview forum: "Tartarus: A Benchmarking Platform for Realistic And Practical Inverse Molecular Design"
_NeurIPS.cc/2023/Track/Datasets_and_Benchmarks — NeurIPS 2023 Datasets and Benchmarks Poster_

### Official Review · Reviewer_8Sdd · 2023-07-16
**An interesting work presenting a benchmark for realistic inverse molecular design, but need significant improvements**

**Rating:** 6
**Confidence:** 4
**Clarity:** Yes.

**Strengths:**

(1) The presented TARTARUS provides four interesting and useful datasets for developing inverse molecule design methods. The tasks of these datasets are tightly related to realistic and practical problems in real-world applications. These datasets can be broadly useful for researchers in the field of inverse molecule design.
(2) The writing of this paper is clear and well-organized. The details of the creating process of all datasets are clearly presented.

**Additional Feedback:**

No additional feedback.

**Correctness:**

The dataset is constructed in a sound way, but several significant baseline methods are not evaluated (see Opportunities For Improvement).

**Documentation:**

Yes.

**Ethics:**

No ethical concerns.

**Limitations:**

Yes.

**Opportunities For Improvement:**

(1) The paper violates the format requirement of 9 page limitation, and does not use the submission template as there are no line numbers.
(2) For baseline methods, this work mainly use string based generation methods, or graph search based methods. However, a large amounts of 2D molecular graph generation methods have been proposed in recent years. Authors are recommended to add at least 1~2 representative baseline methods in this category, e.g., JTVAE [1], GCPN [2], or GraphAF [3].
(3) It seems in all four molecule inverse design tasks, the 3D structures of molecules are needed. However, the benchmark generally takes a procedure of first designing 2D molecular graphs or 1D molecule strings, then compute their 3D conformations. Authors are encouraged to explain or discuss why not directly generating 3D molecules from scratch using a similar way in G-SchNet [4] or EDM [5].
(4) It is not clear to me how molecules are exactly designed. Is it achieved through optimizing given reference molecules or randomly generating by models? Authors are recommended to clarify it.


[1] Junction Tree Variational Autoencoder for Molecular Graph Generation. ICML 2018.
[2] Graph Convolutional Policy Network for Goal-Directed Molecular Graph Generation. NeurIPS 2018.
[3] GraphAF: a Flow-based Autoregressive Model for Molecular Graph Generation. ICLR 2020.
[4] Symmetry-adapted generation of 3D point sets for the targeted discovery of molecules. NeurIPS 2019.
[5] Equivariant Diffusion for Molecule Generation in 3D. ICML 2022.

----Post Rebuttal----
Authors have well addressed all my concerns in the rebuttal. Hence, I raise my score to 6.

**Relation To Prior Work:**

Yes.

**Summary And Contributions:**

This work creates TARTARUS, a benchmark for developing inverse molecular design models. The presented benchmark includes four datasets and tasks. Several moleculae inverse design methods are evaluated on the benchmark.

---

> ### Author Response · Authors · 2023-08-13
>
> Dear Reviewer 4,
>
> We appreciate your detailed feedback and the time you took to review our manuscript. Please find our detailed responses to your comments below:
> 1. The paper violates the format requirement of 9 page limitation, and does not use the submission template as there are no line numbers.
>
> We wish to clarify that the benchmarking and dataset track permits single-blind paper submissions. The 9-page requirement is exclusive to anonymous submissions. An additional page is permitted for non-anonymized submissions, which encompasses all co-authors and an acknowledgements section. Given that we include both, our submission totals 10 pages. Additionally, the fact that the line numbers are missing does not mean that we do not use the submissions template. We do use the submission template. However, we  acknowledge our oversight regarding the missing line numbers and have rectified this in our revised submission.
>
> 2. For baseline methods, this work mainly use string based generation methods, or graph search based methods. However, a large amounts of 2D molecular graph generation methods have been proposed in recent years. Authors are recommended to add at least 1~2 representative baseline methods in this category, e.g., JTVAE [1], GCPN [2], or GraphAF [3].
>
> Thank you for pointing this out. Indeed, there have been a number of new 2D molecular graph generation methods in recent years. Following your recommendation, we are currently working on adding evaluations using representative graph generation methods (e.g., MoFlow [1]), as suggested, to provide a more comprehensive comparison of existing inverse molecular design algorithms. We would like to point out, however, that GB-GA [2] is a “graph-based” generative model, that has demonstrated promising performance on a number of inverse design tasks.
> [1] https://arxiv.org/abs/2006.10137: MoFlow: an invertible flow model for generating molecular graphs.
> [2] Chem. Sci., 2019,10, 3567. https://doi.org/10.1039/C8SC05372C.
>
> 3. It seems in all four molecule inverse design tasks, the 3D structures of molecules are needed. However, the benchmark generally takes a procedure of first designing 2D molecular graphs or 1D molecule strings, then compute their 3D conformations. Authors are encouraged to explain or discuss why not directly generating 3D molecules from scratch using a similar way in G-SchNet [4] or EDM [5].
>
> Thank you for highlighting this point about 3D structures in molecule inverse design tasks. We would like to clarify our rationale:
>
> (a) Our choice of initially designing 2D molecular graphs or 1D molecule strings before computing their 3D conformations was deliberate. While we acknowledge the capabilities of methods like G-SchNet and EDM, many algorithms in the 3D structure generation domain tend to produce a significant number of outliers, affecting the accuracy and reliability of the generated structures. This can lead to significant problems downstream in the simulation pipeline as electronic structure calculations require reasonable geometries to achieve robust convergence.
>
> (b) Additionally, we had to prioritize certain generative models given the scope and focus of our work. That being said, there is no inherent limitation preventing the incorporation of direct 3D molecule generation methods in future iterations of our benchmark.
>
> (c) Finally, the design choice also stems from our intention to create a straightforward comparison across models. Introducing direct 3D generation would complicate the benchmarks, especially since the properties of molecules are intrinsically tied to their 3D structures. In fact, we believe that 3D structure generation would be a benchmark for deep generative models on its own, and looking at molecular properties computed from molecular properties is not necessarily the best metric to evaluate such a benchmark. We believe it would not be meaningful to combine molecular graph and 3D structure generation for a fair comparison against other models that only offer molecular graph generation without specific consideration of 3D structure comparisons. A model that cannot optimize a certain set of properties might just fail to produce sensible 3D structures. By ensuring a uniform 3D structure generation, we can focus on the aspect of molecule generation.
> We hope this clarifies our approach.
>
> 4. It is not clear to me how molecules are exactly designed. Is it achieved through optimizing given reference molecules or randomly generating by models? Authors are recommended to clarify it.
>
> We would like to refer you to the last paragraph of our background section, which states the following: “The model should then propose structures for evaluation by the corresponding benchmark task. Structure optimization is initiated using the best reference molecule from the respective dataset.” Thus, to answer the question of the reviewer, we perform optimization given reference molecules from the corresponding dataset.

---

> > ### Comment · Reviewer_8Sdd · 2023-08-13
> > **Follow-up Responses**
> >
> > I appreciate authors' hard work in the rebuttal.
> >
> > - I know that NeurIPS dataset benchmark allows for both anoymous and non-anoymous submission. However, in my understanding, the 9-page limitation applies to all submission, and I do not find any rules saying that an addition page is allowed for non-anoymous submission. According to the policy ("submission instruction" of [https://nips.cc/Conferences/2023/CallForDatasetsBenchmarks](https://nips.cc/Conferences/2023/CallForDatasetsBenchmarks)), "Submissions are limited to 9 content pages in NeurIPS format, including all figures and tables; additional pages containing the required paper checklist (included in the template), references, and acknowledgements are allowed. If your submission is accepted, you will be allowed an additional content page for the camera-ready version.".  In the current paper, many content of Section 4 appears in Page 10. I think authors should reduce the size of the main body part of paper to fit the 9-page limitation.
> > - I agree with authors' opinion that GB-GA is a graph-based generaton method and 3D molecule generation methods cannot be directly compared with 1D string or 2D graph generation methods. I suggest authors to add the results of new 2D graph generation method such as MoFlow and some 3D molecule generation methods such as G-SchNet or EDM to the revision of paper. Particularly, though authors believe 3D molecule generation methods may not be suitable or perform well for their datasets and tasks, I think it is still necessary to show it by experimental results, and add your analysis in the rebuttal to the experimental results analysis of your paper. As 3D molecule generation is an important topic and your tasks are related to molecular properties that depend on 3D structures, 3D generation methods should not be neglected.
> > - I appreciate authors' clarification that molecules are designed based on given references.
> >
> > As my central concerns about the paper formatting issue and adding 3D molecule generation methods are not completely addressed, I tend to keep my original decision.

---

> > > ### Author Response · Authors · 2023-08-16
> > >
> > > Dear reviewer,
> > >
> > > We thank you for your prompt reply. We have shortened the manuscript, and now our content fits within 9 pages. We hope this addresses any concerns about the length of our submission.
> > >
> > > Regarding the content layout in Section 4, we would like to emphasize the similarity in formatting numerous papers in this track have adopted recently. For instance, [1], the “Outstanding Paper” from last year, predominantly showcases Section 8 on Page 10. Similarly, [2] presents Figure 5 and a significant amount of text on page 10, while [3] features Sections 5.2 and 5.3 on Page 10. These instances should attest to our adherence to a precedent in formatting of NeurIPS submissions.
> > >
> > > [1] https://openreview.net/pdf?id=sde_7ZzGXOE
> > > [2] https://openreview.net/pdf?id=k7FuTOWMOc7
> > > [3] https://openreview.net/pdf?id=zNQBIBKJRkd
> > >
> > > We appreciate your insights regarding the complexities associated with generating 3D conformations of molecules. We would like to emphasize a few points. In order to extend our current benchmarks to properly account for 3D generative models, our entire approach will need significant modifications. First, we would need to provide the corresponding Cartesian coordinates for all the simulations of the provided datasets carried out. This requires a significant additional effort as all the datasets will need to be computed again. Notably, both for the protein ligand design and the chemical reaction substrate design, specialized geometries are required, so it would not be enough to simply train 3D generative models suggesting ground-state gas-phase geometries. Additionally, the chemical reaction substrate design benchmark would not only require one set of Cartesian coordinates, but multiple as both two ground state geometries and one transition state geometry are required. 3D generative models that can properly treat both ground states and transition states are the subject of significant research efforts. This also means that most standard 3D generative models would not be able to fulfill this requirement out of the box, without additional modifications. Moreover, incorporating several new 3D generative models into our work is not just about expanding the scope of models. It is time-intensive as they need proper testing and it necessitates alterations to our property simulation workflows. Hence, we believe that this endeavor, in terms of its sheer amount of required additional work, is far beyond what can be expected from a revision for a NeurIPS submission, but would rather be fitting for a separate benchmark paper, potentially for a future NeurIPS or a similar conference. Accordingly, we believe that while 3D generative models are a very important subject of ongoing research, they are not a simple drop-in replacement for the combination of graph or string generative models paired with dedicated global geometry optimization tools. Additionally, we believe that they also require dedicated benchmarks that are designed to discriminate between the molecular design capability and the 3D structure generation capability. Therefore, overall, we think that they are far outside the scope of what can be reasonably expected from a revision to a submission to NeurIPS.
> > >
> > > Thank you for your constructive feedback. We sincerely hope that our revisions and clarifications address your concerns.

---

> > > > ### Comment · Reviewer_8Sdd · 2023-08-17
> > > > **Response**
> > > >
> > > > I appreciate authors' hard work in addressing my concern about paper length and 3D generation methods. All my concerns are well addressed now so I raise my score to 6. I recommend authors to add these discussions about why not using 3D molecule generation baselines and future plan about them to the revision of paper.

---

> > > > > ### Author Response · Authors · 2023-08-21
> > > > >
> > > > > Dear Reviewer,
> > > > >
> > > > > Thank you once again for your insightful review. We are pleased to inform you that the manuscript now includes results from our implementation of a graph-based flow algorithm, specifically MoFlow. Additionally, we have added data on diversity calculations for each task, which can be accessed in Supplementary Information Tables 2-5. We would be grateful if you could let us know if there are any specific changes that would lead you to consider raising your evaluation score for our manuscript.

---

> ### Author Response · Authors · 2023-08-28
>
> Dear Reviewer,
>
> Thank you once more for your insightful comments on our work. We would like to gently remind you that the rebuttal period ends tomorrow. Should you have any remaining feedback or need further clarification on our revisions or responses, please don't hesitate to reach out. We are more than willing to address any remaining concerns.
>
> Warm regards, Authors

---

### Official Review · Reviewer_yzPs · 2023-07-21
**Review of "Tartarus: A Benchmarking Platform for Realistic And Practical Inverse Molecular Design"**

**Rating:** 6
**Confidence:** 4
**Correctness:** Yes.
**Clarity:** Yes.

**Strengths:**

The proposed evaluation methods based on physical simulation are all sound. And the evaluation is aligned of the real-world application.
The analysis of some selected molecule design algorithms reflects that there is still a lot of root for developing more useful methods.
The codebase of this benchmark is well-developed, which provides a good standard to follow up for researchers in this community.

**Additional Feedback:**

I recommend the authors to design a benchmark for one molecule design task (e.g., design of chemical reaction substrates), develop a more dedicated evaluation workflow for this single task, and make more in-depth analysis of the existing algorithms instead of putting many tasks in one benchmark due to the specification I mentioned above.

**Documentation:**

Yes.

**Ethics:**

No.

**Limitations:**

Yes.

**Opportunities For Improvement:**

This work tries to propose a benchmark for several molecule design tasks. However, I think it would be better to propose a benchmark for one molecule design task, respectively. My motivation is that, for example, researchers in material design and drug discovery may develop specific algorithms to design molecules. This specification implies that it may be somehow meaningless to try to propose a unified benchmark.

The experiments are far from comprehensive. Some representative molecule design algorithms are missing.

**Relation To Prior Work:**

Some previously used evaluation methods are not mentioned. For example, for designing protein-ligand, there are already some widely-used similar evaluation methods based on physical simulations in the community of AI for drug discovery.

**Summary And Contributions:**

This paper provides benchmark tasks based on physical simulation or molecular systems. The benchmark tasks include designing organic photovoltaics, organic emitters, protein ligands, and chemical reaction substrates. Detailed descriptions and implementation of the evaluation workflow are provided. The proposed benchmark reflects actual problems in chemistry and provides new opportunities for developing more molecule design algorithms, especially generative models. In summary, this benchmark provided standard evaluation workflows based on physical simulation that is more aligned with the real-world application.

---

> ### Author Response · Authors · 2023-08-13
>
> Dear Reviewer 3,
>
> We greatly appreciate the time and effort you invested in reviewing our manuscript. Your feedback is invaluable to us as it helps us to improve it. Here are our detailed responses to your comments:
>
> Unified Benchmark for Multiple Molecule Design Tasks:
> The essence of our unified benchmark suite is to lay a foundational structure that the community can expand upon. While we provide a holistic framework, it is designed with modularity in mind. Thus, researchers can select specific benchmarks from our suite aligning most closely with their applications, ensuring that the results are both relevant and specific to the target design problem. Hence, we believe that we did follow your recommendation to design a benchmark for one molecule design task, we just did this four times. We also agree with your point that specialized benchmarks are needed, but it is worth noting that specialized benchmarks, while valuable for the intended application domain, are prone to introduce significant biases for developing new models. This phenomenon is particularly evident in important domains like drug discovery, where many specialized generative models are published without a comprehensive consideration of their domain specificity. Nevertheless, largely unfounded claims about their generalizability across multiple domains are still made. We believe that our platform will change this and provide researchers with an effective means to assess these claims regarding generalizability.
>
> Comprehensiveness of Experiments:
> We acknowledge that the selection of models in our initial experiments could be more comprehensive. Nevertheless, we would like to mention that we already have a graph-based molecular design algorithm with GB-GA. Following your recommendation, we are currently working on incorporating evaluations using additional graph methods (e.g., MoFlow [1]) to offer a broader comparative landscape. However, we would like to point out that we believe developing, implementing, and running all the benchmarks of TARTARUS was comprehensive. The sheer amount of numbers in the tables may not necessarily appear as much, but what is behind each result is a substantial amount of work. Already implementing and testing all the simulation workflows for the benchmarks alone is a significant effort, especially since these types of workflows are not standardized in the literature for these types of design problems. While the selection of molecular design algorithms could still be more comprehensive, we believe it covers many important classes of models.
>
> [1] https://arxiv.org/abs/2006.10137: MoFlow: an invertible flow model for generating molecular graphs." In Proceedings of the 26th ACM SIGKDD International Conference on Knowledge Discovery & Data Mining, pp. 617-626. 2020.
>
> Relation to Prior Work:
> We thank you for this important remark. To follow your recommendation, we added several recent publications demonstrating the use of molecular docking for benchmarking deep generative models.
>
> Overall, your feedback has been instrumental in refining both our manuscript and approach. We strive to ensure our work is both comprehensive and adaptable for the community, and we are eager to address any further concerns or insights you might have.

---

> > ### Comment · Reviewer_yzPs · 2023-08-20
> >
> > Thanks for your rebuttal!
> >
> > Your response to my concern about the unified benchmark is reasonable. I acknowledge that many generative models are specialized for their domain, and may not work in other domains. From this perspective, this benchmark provides a better way to overall assess the molecule generative models in four downstream tasks. And the methods you have tested in the benchmark are general in that sense. In other words, the selected molecule design methods can be easily adopted in different downstream tasks. I think this may be the main point of this work. **I hope the authors emphasize this point in the revised paper.** This also addresses my concern about the comprehensiveness of experiments. There are many works that design protein ligands using ML. However, some of them can not work directly in other scenarios, such as chemical reaction substrates. So I have re-evaluated the comprehensiveness of the experiments and do not regard this as a weakness of this work.
> >
> > Design general and specialized methods are both meaningful, respectively. I would keep my point that a specialized benchmark for a certain task is still needed because some problems of a certain domain may be critical in practice and may not be shared with other domains. And such a benchmark may be designed more carefully and be more valuable especially when a certain application is considered. For example, such as designing protein ligands, the methods that are tested in this work mostly fall into the category of ligand-based drug design (LBDD) [1]. However, there are other categories, such as structure-based drug design (SBDD), such as [2,3,4,5,6,7]. This should be mentioned and clarified that these methods are specially proposed for designing protein ligands and may not be adapted to other tasks. I think this may be a proper reason that you do not test these methods in experiments of this benchmark.
> >
> > In summary, I hope the authors clarify and emphasize that **this benchmark is designed for evaluating **general** molecule design methods and can not replace the benchmarks that are specially designed for a certain task**, such as designing chemical reaction substrates. From this perspective, if generality is the main point of this benchmark, four downstream tasks may be too few. There are many other molecule design tasks that can be taken into consideration, such as linker design and scaffold hoping. I look forward to the response of the authors to my follow-up comments. I would consider raising my rating then.
> >
> > **References**
> > [1] Aparoy, Polamarasetty, Kakularam Kumar Reddy, and Pallu Reddanna. "Structure and ligand based drug design strategies in the development of novel 5-LOX inhibitors." Current medicinal chemistry 2012
> > [2] Peng, Xingang, et al. "Pocket2mol: Efficient molecular sampling based on 3d protein pockets." ICML 2022
> > [3] Schneuing, Arne, et al. "Structure-based drug design with equivariant diffusion models." arXiv preprint 2022
> > [4] Guan, Jiaqi, et al. "3d equivariant diffusion for target-aware molecule generation and affinity prediction." ICLR 2023
> > [5] Zhang, Zaixi, et al. "Molecule generation for target protein binding with structural motifs." ICLR 2023
> > [6] Guan, Jiaqi, et al. "DecompDiff: Diffusion Models with Decomposed Priors for Structure-Based Drug Design." ICML 2023
> > [7] Zhang, Zaixi, and Qi Liu. "Learning Subpocket Prototypes for Generalizable Structure-based Drug Design." ICML 2023

---

> ### Comment · Reviewer_yzPs · 2023-08-21
>
> After careful consideration, I decided to raise my score to 6.
>
> It is no doubt that the authors have put a lot of effort into this work. To further improve this work, I recommend the authors discuss the significance of this benchmark from the perspective of generality and specification. The specified benchmark for a specific task is still needed in the area. Nevertheless, this work can be viewed as a step towards benchmarking general machine-learning-based molecule design methods. I hope the authors acknowledge the limitations I mentioned above and objectively discuss them in the revised paper.
>
> I sincerely hope this benchmark can play an important and fundamental role in motivating researchers to develop more practical inverse molecule design methods. I also hope the authors can keep updating this work, i.e., putting more inverse molecule design tasks in the benchmark in the future.

---

> > ### Author Response · Authors · 2023-08-21
> >
> > Dear Reviewer 3,
> >
> > We thank you for emphasizing this important distinction between general and specific molecular design. We agree with your point that developing benchmark tasks very specific to a particular domain of interest are as important as having a more general molecular design framework. Accordingly, we emphasized the focus on general molecular design already in the introduction: “The core idea is to provide a unified benchmarking framework with a diverse selection of problem domains to assess the generalizability of inverse design algorithms.” Additionally, we agree with you that four benchmark domains is not enough to assess generality but we think Tartarus provides an important step towards generality. We are definitely planning to expand the current set of benchmarks to other domains. Accordingly, we added the following sentence to the introduction: “Notably, the scope of included benchmark domains is to be expanded upon in future work.” We also repeated the focus on generality in the conclusion section: “We observed that none of the representative generative models selected consistently outperformed the others across all the design tasks. This is due to a seemingly strong dependence of the performance of some models on the specific benchmark domain and suggests the importance of conducting diverse molecular design benchmarks that reflect actual problems pursued in chemistry to evaluate the generalizability of inverse molecular design algorithms across domains.”
> >
> > Additionally, we also added a brief discussion about general versus specific molecular design to the conclusion section: “Notably, we would like to emphasize that Tartarus does not replace the development of more specialized molecular design benchmarks that are tailored to specific domains of interest and necessarily follow a different design workflow, particularly in drug design. We believe that such specialized benchmarks are complementary to Tartarus and are equally important to assess the practicality and performance of molecular generative models.“ We hope that this addresses your points adequately. We would like to thank you again for your insightful feedback that allowed us to improve our manuscript significantly. We would be grateful if you could let us know if there are any specific changes that would lead you to consider raising your evaluation score for our manuscript.

---

> ### Author Response · Authors · 2023-08-28
>
> Dear Reviewer,
>
> Thank you once more for your insightful comments on our work. We would like to gently remind you that the rebuttal period ends tomorrow. Should you have any remaining feedback or need further clarification on our revisions or responses, please don't hesitate to reach out. We are more than willing to address any remaining concerns.
>
> Warm regards, Authors

---

### Official Review · Reviewer_zwfp · 2023-07-23
**A new realistic benchmarking platform for molecular inverse design**

**Rating:** 7
**Confidence:** 5
**Correctness:** Both the simulation methods and evalu…
**Clarity:** The paper is clear, coherent, and wel…

**Strengths:**

- The paper develops a simulation-based workflow to evaluate generated molecules, employing openbabel, Crest, XTB, etc. These evaluation workflows are much more robust than widely used LogP, synthesizability scores, etc.
- The benchmark tasks cover a broad range of scientific domains, including organic photovoltaic design, protein ligand design, chemical reaction subtract design.
- It also extensively benchmarked 7 different molecular generative model algorithm


**Additional Feedback:**

No additional feedback.


**Documentation:**

The data is available on github. There is enough detail on data collection methods. The detailed setting to reproduce benchmark results are available at supplementary information.

**Ethics:**

No ethics concerns.

**Limitations:**

The authors didn’t adequately discuss the negative social impact of their work. For example, the molecular generative models might be used to generate chemical weapons and hazardous materials.

**Opportunities For Improvement:**

- The benchmarked generative model algorithms are all based on SMILES and SELFIES representations. It would be great to see graph-based methods like JT-VAE and 3D molecular generative methods like diffusion models to be included in the benchmark
- The benchmark does not include diversity metrics, which is important for evaluating generative models.


**Relation To Prior Work:**

The authors discussed how their work differs from previous contributions in the background section.


**Summary And Contributions:**

The paper presents a new molecular inverse design benchmark featuring 4 newly designed realistic tasks. For each task, it provides training datasets, simulation workflows to evaluate generated molecules, and benchmarked several molecular generative models. It is a timely benchmark platform to provide realistic tasks for evaluating molecular generative models.

---

> ### Author Response · Authors · 2023-08-13
>
> Dear Reviewer 2,
>
> Thank you for your comprehensive review and positive feedback for our manuscript. We are glad to hear that you found our benchmark platform timely and relevant. Please find our detailed responses to your comments below:
>
> Strengths:
> We appreciate your recognition of the robustness of our evaluation workflows, the breadth of our benchmark tasks, and our extensive benchmarking of various molecular generative models.
>
> Opportunities For Improvement:
> 1. Generative Model Algorithms and Representations:
> We acknowledge your suggestion to expand the scope of our benchmarking and include graph-based methods like JT-VAE and 3D molecular generative methods such as diffusion models. We would like to note that with GB-GA we already have provided results for a graph-based method. Additionally, we do not believe that using a 3D molecular generative model allows a fair comparison between the molecular design algorithms, unless the Cartesian coordinates generated by the model are ignored and new ones are found by our simulation workflow. This is because the properties of our molecular design benchmarks depend on these Cartesian coordinates. Hence, comparison of the final molecular properties would incorporate both a geometric and a structure design component. Thus, we believe that separate benchmarks that compare the 3D structure generated would be necessary to properly benchmark 3D molecular generative models. Nevertheless, regarding alternative graph-based methods, we appreciate the suggestion of the reviewer as their inclusion would indeed provide a more comprehensive view of the landscape, and we are currently working on implementing and running such approaches.
>
> 2. Diversity Metrics:
> We thank you for pointing out the importance of diversity metrics in evaluating generative models. Following this suggestion, we are currently working on implementing diversity metrics in our evaluations, providing a more rounded assessment of the capabilities of the generative models tested.
>
> 3. Social Impact:
> Your point on the potential negative societal implications of our work is well taken. We understand the gravity of these concerns, especially in the context of possible misuse of molecular generative models. Thus, we have incorporated an entire paragraph about this issue into the Background section of our manuscript: “Molecular design algorithms have the potential to accelerate the discovery of materials for the benefit of humankind. However, they can just as well be used to design hazardous materials intentionally. While this pertains primarily to the molecular design algorithms themselves, benchmarking can also provide a contribution in that regard by identifying robust and generally applicable tools. However, first, we would like to emphasize that none of the molecular design benchmarks developed have a direct link to the development of potentially hazardous materials. Additionally, the molecular design algorithms employed in this work propose merely structures but do not provide any instructions for their synthesis making their malicious use unlikely.”

---

> > ### Author Response · Authors · 2023-08-21
> >
> > Dear Reviewer,
> >
> > Thank you once again for your insightful review. We are pleased to inform you that the manuscript now includes results from our implementation of a graph-based flow algorithm, specifically MoFlow. Additionally, we have added data on diversity calculations for each task, which can be accessed in Supplementary Information Tables 2-5. We would be grateful if you could let us know if there are any specific changes that would lead you to consider raising your evaluation score for our manuscript.

---

> ### Author Response · Authors · 2023-08-28
>
> Dear Reviewer,
>
> Thank you once more for your insightful comments on our work. We would like to gently remind you that the rebuttal period ends tomorrow. Should you have any remaining feedback or need further clarification on our revisions or responses, please don't hesitate to reach out. We are more than willing to address any remaining concerns.
>
> Warm regards, Authors

---

### Official Review · Reviewer_uBzk · 2023-07-25
**Tartarus - benchmarking**

**Rating:** 6
**Confidence:** 4
**Correctness:** Correct.

**Strengths:**

In a time when paradox of choice is gripping every domain, what kind of deep generative model may become suitable for a given task is an important problem. Authors have competently executed their to address this important problem and came to the conclusion that the degree of sophistication of the deep generative model (or the manner in which chemical featurization/encoding is carried out) does not necessarily guarantee stronger performance in inverse design problems.

**Additional Feedback:**

Nothing to add

**Clarity:**

The take home message is not very clear. Readout from various tables should have been more explicit.

**Documentation:**

Yes

**Ethics:**

No concerns

**Limitations:**

I don’t see much effort to rationalize what could have been the origin of performance differences between different generation tasks. While these models (VAE) are inherently complex, their relative strengths and weakness should be commented to get a better feel of the trends reported through different tables in the manuscript.

**Opportunities For Improvement:**

Comments on how molecular property (HOMO-LUMO gap, LUMO energy and dipole moment, as used in this study) translates to that of the corresponding photovoltaic material would add value. For instance, a quality such as dipole moment depends on the mode of measurement as well as on what state of matter the molecule exists. This further implies that packing might as well lead to no net (measurable) dipole moment. This comment has implications to additional assumptions, such as searching of lowest energy conformer and the energy obtained through single point calculation, on which the promise that the study would of use to photovoltaic materials.

Authors should use standard terminologies for mean and std.dev., instead of additional notations in a domain which is already flooded with non-standard terms and terminologies to mean what individual authors want them to mean.

Several places in the manuscript authors say that the generated entities were either comparable or better than the best molecules in the training dataset. This requires additional clarification as to how many training set samples were indeed known experimentally. How good was the validation in comparison to known (limited?) experimental values?


**Relation To Prior Work:**

A lot of their models were already available, so are the datasets.
Although work is good, it is not scoring well on novelty. Most of what they have reported here is based on previous works on OSC/OPVs (cited refs. 31, 34, 37 etc.,) and generative tasks (ref. 28). With access to various datasets, and previous ‘baseline’ models, it is not very clear what were the new components, except a few tables bearing numbers (with no scholarly efforts to comment on these numbers)

**Summary And Contributions:**

The study aims to propose realistic benchmarks for complex molecular design problems. The data has been generated using suitably positioned simulation as practiced in the domain of molecular materials, drugs, and chemical reactions. The proposed method is a conglomeration of modules, each dedicated for a particular task (ranging from conformer generation to electronic structure computations, HOMO-LUMO energies, singlet-triplet gap, docking, and so on), set to generate molecules of potential interest to photovoltaics, LEDs, and drug design. Key molecular properties such as power conversion efficiency, blue light emission potential as gauged using minimal S-T gap and maximum oscillator strength, docking score for drug efficacy were being optimized under the deep generative settings.

---

> ### Author Response · Authors · 2023-08-13
>
> Dear Reviewer 1,
>
> Thank you for your review. Please see our detailed comments below.
>
> Strengths:
> We appreciate your acknowledgment of the strengths of our work and your summary of our findings.
>
> Opportunities For Improvement:
> 1. Molecular Properties:
> We strongly appreciate your important comment and we fully agree. In lieu of the strict page limit for submissions, we already have a comprehensive discussion of these points for every set of benchmark tasks in the Supporting Information, sections S3.A-D. In particular, we outline how the simulation results are used to compute the benchmark metrics and the corresponding  underlying assumptions. Additionally, we provide relevant references for key models incorporated in the corresponding simulation workflows. We believe that these points are better raised in a dedicated section in the Supporting Information rather than adding separate remarks without context to the main text. To refer the reader to this section, we added the following sentence to the first paragraph of the Results: “For a comprehensive outline and discussion of the property simulation workflows developed in this work, we refer the reader to the Additional Results of the Supporting Information.”
> 2. Standard Terminologies:
> We thank you for pointing this out. To follow your recommendation, we removed the non-standard notation regarding mean and standard deviation from all the tables.
> 3. Experimental Validation:
> While we appreciate this important remark, we would like to emphasize that this point is beyond the scope of this work. The idea of our work is to provide a benchmarking framework based on established approaches to simulate molecular properties of interest in domains where simulations have been used comprehensively before, and thus validated. However, the molecule datasets were not selected with respect to the availability of experimental data. To clarify this point, we added the following sentence to the last paragraph of the Background section: “Notably, the potential availability of reference data from laboratory experiments for at least subsets of our curated datasets was not considered.”
>
> Limitations:
> Thank you for your feedback. We recognize that determining the exact origin of performance differences wasn't our central focus. We've elaborated on general model characteristics in the Supporting Information. Additionally, we added the following sentence to the Conclusion and Outlook: “We note that relatively simple approaches like genetic algorithms that do not need large reference datasets show more consistently good performance across benchmarks, whereas the VAE approaches, that strongly rely on the available training data, show limited performance across multiple benchmarks. However, more comprehensive investigations are required to confirm this hypothesis and identify the origins of the limited performance of some of the approaches tested.” Our work marks the start of a deeper dive into these performance disparities, and we believe our benchmark will aid in this exploration.
>
> Clarity:
> To make the take-home message clearer, we added the following sentence to the Abstract: “Surprisingly, we find that model performance can be strongly dependent on the benchmark domain.” Additionally, we added the following sentence to the Conclusion and Outlook: “This is due to a seemingly strong dependence of the performance of some of the models on the specific benchmark task and domain.” Additionally, we made readout from all the benchmark tables more explicit by directly referencing the numbers in the tables to support our statements. Furthermore, we decided to underline the second-best benchmark results in each table to make readout more reader friendly.
>
> Relation To Prior Work:
> While we agree with you that the molecular design algorithms were already available, we strongly disagree that the datasets were available before and hence are not novel. The datasets consist of the molecules, in the form of SMILES, and their predicted properties. For the organic photovoltaic and protein ligand benchmarks, though molecules were previously published, our predicted properties are based on fresh computational workflows detailed in this paper. Significant attention was given to the property simulation workflows in the Supporting Information. We optimized methods for Cartesian coordinate generation, electronic structure simulation, and performance metric estimation, emphasizing code robustness. We revisited properties for the Harvard Clean Energy subset and introduced new ones for the DTP Open Compound collection. The GDB-13 subset we used is unprecedented and tailored for this domain using unique filters. Our predicted molecular properties are novel. The chemical reaction substrate optimization benchmark presents a brand new dataset and simulation method. We believe our efforts in benchmark simulation workflows and datasets offer significant contributions to the community.

---

> > ### Author Response · Authors · 2023-08-21
> >
> > Dear Reviewer,
> >
> > Thank you once again for your insightful review. We are pleased to inform you that the manuscript now includes results from our implementation of a graph-based flow algorithm, specifically MoFlow. Additionally, we have added data on diversity calculations for each task, which can be accessed in Supplementary Information Tables 2-5. We would be grateful if you could let us know if there are any specific changes that would lead you to consider raising your evaluation score for our manuscript.

---

> > > ### Author Response · Authors · 2023-08-28
> > >
> > > Dear Reviewer,
> > >
> > > Thank you once more for your insightful comments on our work. We would like to gently remind you that the rebuttal period ends tomorrow. Should you have any remaining feedback or need further clarification on our revisions or responses, please don't hesitate to reach out. We are more than willing to address any remaining concerns.
> > >
> > > Warm regards, Authors

---

### Comment · Area_Chair_hh7z · 2023-08-17
**Please respond to rebuttals!**

Hi reviewers,

Thank you for your hard work writing reviews.The authors have written some pretty extensive rebuttals. Please at least acknowledge the rebuttals. Ideally, reply with any outstanding concerns and a justification for keeping or raising your score. This helps the authors improve their work, and will help me with decisions and metareviews.

PS if the authors addressed all your concerns, it is hard for me to understand a score lower than 6.

---

### Decision · Program_Chairs · 2023-09-22

**Decision:**

Accept (Poster)

**Comment:**

## Summary

The authors curate training sets and develop simulation-based evaluation suites for several interesting generative molecular design problems. They then evaluate a set of generative models.

## Strengths

- [quality] The codebase of this benchmark is well-developed, which provides a good standard to follow up for researchers in this community.
- [significance] The analysis of selected molecule design algorithms reflects that there is still a lot of root for developing more useful methods.
- [quality] The proposed evaluation methods based on physical simulations are all sound. And the evaluation is aligned of the real-world application.
- [significance] The benchmark tasks cover a broad range of important scientific domains, including organic photovoltaic design, protein ligand design, chemical reaction substrate design.
- [clarity] The paper is very easy to read.

## Weaknesses

- [quality] Not comparing to experimental values weakens the conclusions